# Clustering induces switching between phoretic and osmotic propulsion in active colloidal rafts

Dolachai Boniface [1,3], Sergi G. Leyva [1,2,3], Ignacio Pagonabarraga [1,2] & Pietro Tierno [1,2] ✉

Active particles driven by chemical reactions are the subject of intense research to date due to their rich physics, being intrinsically far from equilibrium, and their multiple technological applications. Recent attention in this field is now shifting towards exploring the fascinating dynamics of active and passive mixtures. Here we realize active colloidal rafts, composed of a single catalytic particle encircled by several shells of passive microspheres, and assembled via light-activated chemophoresis. We show that the cluster propulsion mechanism transits from diffusiophoretic to diffusioosmotic as the number of colloidal shells increases. Using the Lorentz reciprocal theorem, we demonstrate that in large clusters self-propulsion emerges by considering the hydrodynamic flow via the diffusioosmotic response of the substrate. The dynamics in our active colloidal rafts are governed by the interplay between phoretic and osmotic effects. Thus, our work highlights their importance in understanding the rich physics of active catalytic systems.

In the past few years, active colloidal particles have led to several exciting developments in the field of non-equilibrium statistical mechanics[1–4] while being used as simplified models to reproduce emerging phenomena in biological self-propelling systems[5–8]. Since the pioneering works of Ismagilov et al.[9] and Paxton et al.[10], chemical reactions have been routinely used to induce propulsion in asymmetric systems[11], including Janus particles[12–15], nanorods[16,17], dimers[18,19], mixtures[20,21] and many others[22–24]. Besides the interest in the reaction mechanism that leads to the net motion, these particles have shown the capabilities to pick up, transport, and release microscopic cargoes on command[25–28]. Thus, active particles may find direct applications in different technological fields, including biomedicine[29], targeted drug delivery[30] and microfluidics[31].

In several catalytic systems, self-propulsion is typically attributed to electrophoresis[32] and/or chemophoresis[33], namely the motion of particles in an electric field/concentration gradient generated by the chemical activity of the particles[34]. However, the release or consumption of chemical elements also generates a concentration gradient along the surfaces in the proximity of a catalytic system. Since most of the self-propelled catalytic systems evolve close to a substrate, one can expect the presence of a local osmotic flow, which may affect the system dynamics through hydrodynamic interactions[35]. Moreover, the osmotic flows on the substrate may even counteract and compete with particle diffusiophoresis. This competition has already been exploited to concentrate passive nanoparticles in a capillary channel[36]. Due to their similar origin[37], the contributions of diffusiophoresis and substrate diffusioosmosis on the self-propulsion of active colloids are challenging to disentangle[33]. As a consequence, most of the theoretical and simulation models in the field do not consider the impact of hydrodynamic interactions associated with substrate diffusioosmosis, and often employ an "ad hoc", effective diffusiophoresis to describe the experimental results. In contrast, a recent theoretical work has demonstrated that the diffusioosmotic contribution can even guide active Janus particles across a chemically patterned substrate[38].

Here, we combine experiments and theory to demonstrate that the diffusioosmotic flow induced by a catalytic particle due to a near surface is necessary to describe the motion of active clusters driven by chemical reactions. We realize photoactivated colloidal rafts

[1]Departament de Física de la Matèria Condensada, Universitat de Barcelona, 08028 Barcelona, Spain. [2]University of Barcelona Institute of Complex Systems (UBICS), 08028 Barcelona, Spain. [3]These authors contributed equally: Dolachai Boniface, Sergi G. Leyva. ✉e-mail: ptierno@ub.edu

composed of multiple shells of passive spheres around a single apolar catalytic particle, and investigate the raft kinetics and dynamics during the illumination process. These clusters grow up to an area of 120 times the silica colloids, corresponding to 7 compact shells of passive spheres. We find that the clusters display self-propulsion even though they consist of symmetric shells of passive spheres. We thus realize a singular catalytic self-propelled system, characterized by an evolving shape, with an aspect ratio gradually approaching that of a flat disk. Numerical simulations based only on a purely diffusiophoretic system, without osmotic flow on the substrate, reproduce the raft kinetics but not the cluster direction of motion and its persistence length for large clusters. We show that the substrate osmotic flow is an essential feature to explain the mechanism of motion of these composite clusters.

## Results

### Colloidal rafts

Our colloidal rafts are realized by illuminating with blue light (wavelength $\lambda = 450 - 490$ nm) synthesized ellipsoidal particles made of hematite and characterized by a short and a long axis equal to 1.3 μm and 1.8 μm resp., Fig. 1(A). These particles are dispersed with passive silica spheres (1 μm diameter) in an aqueous solution of hydrogen peroxide (3.6% w/v). The pH of the solution is raised to ~ 9.2 by adding Trimethylphenylammonium (TMAH) to negatively charge the surfaces[39]. The electrostatic repulsion stabilizes the dispersion and prevents the colloids from sticking to the substrate. The colloidal dispersion is sedimented over the bottom of a sealed rectangular capillary tube, such that the particles are 100 μm far away from the top plate. We measure independently the diffusion coefficient of the hematite and silica colloids, respectively $D_a = 0.16$ μm² s⁻¹ and $D_p = 0.29$ μm² s⁻¹, see Supplementary Fig. 4 and Supplementary Note 1. From $D_a$ and using the Faxen's result for the drag of a sphere near a single wall[40], we estimate the average distance between the substrate and the surface of the hematite particle to be $h \sim 0.3$ μm. The relative density is below 1 active particle for every 2000 passive ones, with a total surface fraction of ~6%. More details on the experimental protocol are given in the Materials and Method section and in the Supplementary Information file including an image of the experimental setup (Supplementary Fig. 1) and additional details in Supplementary Note 1.

Once the light is applied, the hematite particle initiates the decomposition of hydrogen peroxide in water, following the chemical reaction: $2H_2O_{2(l)} \rightarrow O_{2(g)} + 2H_2O_{(l)}$. As a consequence of this reaction, the particle becomes active and induces a strong phoretic attraction of the passive spheres, which assemble them in the form of a circular cluster as depicted in Fig. 1B. During growth, the raft translates and rotates, displaying looping trajectories, Fig. 1C and Supplementary Movie 1. The self-assembly process can be completely and reversibly controlled by the light intensity, as shown in the sequence of images in, Fig. 1D where one large cluster is disassembled by a step-wise reduction of the light power. At the maximum intensity of $I = 125$ mW cm⁻², one hematite accumulates up to 6–7 layers of passive particles, i.e. more than 100 colloids, for a one-hour experiment. In contrast, at the minimum light intensity, which is ~ 28 times lower, there is only one layer. To obtain enough statistics, these experiments were repeated for 12 times.

As shown in Fig. 1E, the cluster growth is characterized by a simple logarithmic relationship. Furthermore, we find that the rafts follow a sub-linear growth with a power law behavior up to $t = 2000$ s ($\simeq 0.6$ h), inset in Fig. 1F. The exponent 1/3 is consistent with the Ostwald coarsening process, as described by the Lifshitz-Slyozov-Wagner theory[41]. Such an exponent was predicted in the scalar field theory of active systems[42] and recently observed in experiments with clusters of passive particles induced by active agents[43]. The mean cluster velocity $\bar{v}_c$ decreases linearly with the cluster area $A$, reducing almost to zero for the largest size of $A = 175$ μm², as shown in the top inset in Fig. 1F. We note that the raft formation can only be obtained due to the phoretic flow, induced by the photoactivated decomposition of hydrogen peroxide in water. Indeed, in a separate set of experiments, we have checked that in absence of light (Supplementary Movie 2) or in absence of hydrogen peroxide but under blue light (Supplementary Movie 3) both the hematite and the silica particles display simple diffusive dynamics without any signs of phoretic attraction.

The aggregation process arises from chemophoresis, induced by the concentration gradient generated by the hematite particle[44]. The saturation of the raft size corresponds to a balance between the effective phoretic potential energy and the thermal energy, the latter

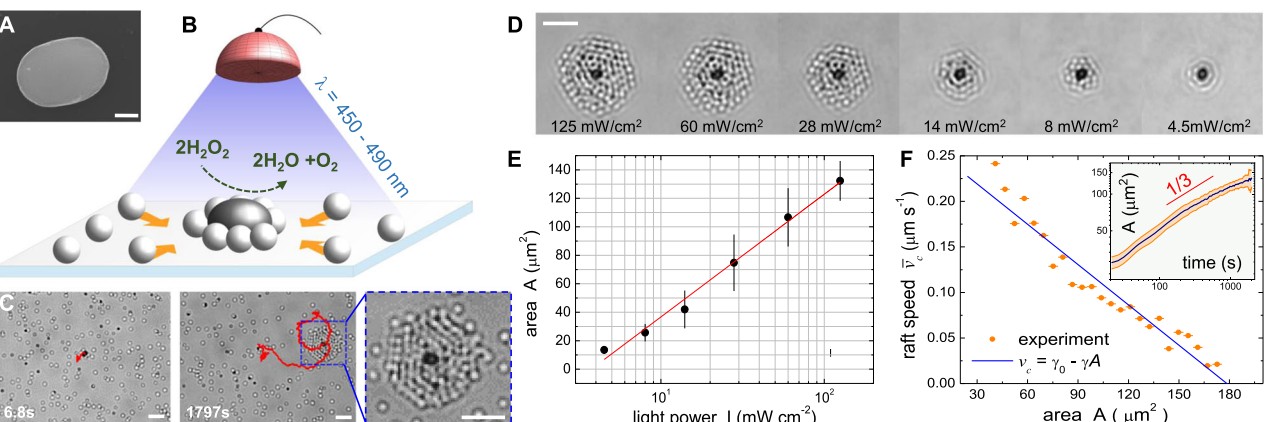

**Fig. 1 | The active colloidal raft. A** Electron microscopy image of one hematite particle, scale bar is 500 nm. **B** Schematic showing the assembly of the colloidal raft under blue light in a water ($H_2O$) hydrogen peroxide ($H_2O_2$) mixture. **C** Sequence of three optical microscope images of a growing raft with superimposed the trajectory of the central active particle (red line). Time $t = 0$s corresponds to light application. Scale bar is 5 μm for all images. The last image displays the final cluster size, see Supplementary Movie 1. **D** Sequence of images showing the steady state area occupied by the cluster at different light intensities, the scale bar is 5 μm. **E** Cluster area $A$ versus light intensity $I$. Black dots are the average values from 5

different experiments. The error bars are the confidence interval for $P = 0.95$. The straight red line is a linear regression using the logarithm of the light intensity, $A(I) = A_0 \ln(I/I_0)$ with $A_0 = 37.4 \pm 1.9$ μm² and $I_0 = 3.74 \pm 0.4$ mW cm⁻². **F** Average raft velocity $\bar{v}_c$ versus cluster area $A$ showing the experimental data (orange disks) and a linear regression with $\gamma_0 = 0.26 \pm 0.02$ μms⁻¹ and a negative slope $\gamma = (1.48 \pm 0.02) \cdot 10^{-3}$ μm⁻¹ s⁻¹. Inset shows a log-log plot of the area versus time for several rafts, error bars are indicated by the shaded orange region. In both plots errors result from the statistical average of different measurements.

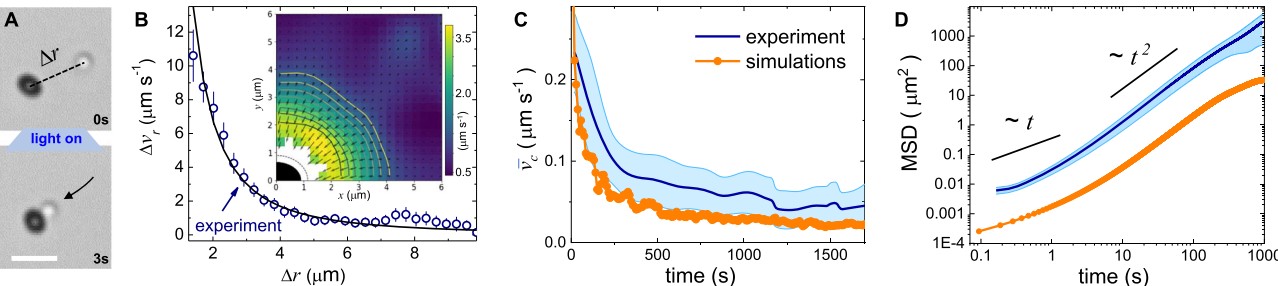

**Fig. 2 | Raft dynamics. A** Sequence of images showing the attraction of a silica colloid towards the hematite particle once blue light is applied ($t = 0$). **B** Relative speed $\Delta v_r$ versus relative distance $\Delta r$: the solid line is fitted to the data (blue circles) following Eq. (3). Inset displays a heat map of the velocity and direction of a passive particle near the hematite one. The errors in the graph have been obtained from the statistical average of 64 independent experiments. **C** Mean cluster speed $\bar{v}_c$ and **D** mean square displacement versus time from experiments (blue lines) and numerical simulations (orange disks). In both graphs the shaded blue regions denote the confidence interval, indicating the range within which the averages (blue lines) are with a probability of 95%, assuming a Student's t-distribution.

of the order of $k_B T$, being $k_B$ the Boltzmann constant and $T \sim 293 K$ the thermodynamic temperature. However, the emergence of self-propulsion in our system is more subtle and unexpected. Indeed, it was demonstrated that the decomposition of hydrogen peroxide in water induces propulsion in Janus colloids with anisotropic coating[45,46]. However, our clusters appear symmetrically surrounded by shells of passive spheres, a situation that would in principle preclude the emergence of directed motion unless some spatial symmetries in the system are broken.

## Numerical simulations

To understand the kinetics and the self-propulsion behavior of the rafts, we first perform Brownian dynamic simulations using input parameters directly obtained from the experimental pair interaction between the hematite and the silica colloids. We model a purely diffusiophoretic system consisting a bath of $i = 1 .. N$ passive particles at positions $\boldsymbol{R}_i$ (diameter $\sigma_p = 1 \mu m$, surface mobility $\mu_p$ and diffusion coefficient $D_p$) with a single active particle. To reproduce the aspect ratio of the experimental ellipsoids, we modeled the hematite as a dumbbell consisting of two active particles, $\alpha = 1, 2$, located at positions $\boldsymbol{r}_\alpha$ (diameter $\sigma_a = 1.3 \mu m$, surface mobility $\mu_a$, and diffusion coefficient $D_a$) and kept by a spring with rest length $l_0 = 0.5 \mu m$, and force of magnitude $F^h$ along the vector $\hat{\boldsymbol{n}}_i = (\boldsymbol{r}_i - \boldsymbol{r}_j)/r_{ij}$ joining the two beads. Accordingly, the equation of motion of the active and passive colloids read:

$$\dot{\boldsymbol{r}}_\alpha = \boldsymbol{v}_\alpha + (F^h \hat{\boldsymbol{n}}_\alpha + \boldsymbol{F}^c_\alpha)/\gamma_a + \sqrt{2D_a}\boldsymbol{\xi}_\alpha, \tag{1}$$

$$\dot{\boldsymbol{R}}_i = \boldsymbol{V}_i + \boldsymbol{F}^c_i/\gamma_p + \sqrt{2D_p}\boldsymbol{\xi}_i. \tag{2}$$

where $\gamma_a$ and $\gamma_p$ correspond to the friction coefficients of the active and passive particles, respectively, while $\boldsymbol{F}^c_i$ and $\boldsymbol{F}^c_\alpha$ account for the steric forces given by a Weeks-Chandler-Andersen potential, which prevents particles from overlapping. The term $\boldsymbol{\xi}_i$ is a random Gaussian noise that accounts for the thermal bath. Each bead constituting the dumbbell in the hematite acts as a source[22,44,47] of a chemical field, $\phi$. We note here that the diffusion coefficients of dioxygen and hydrogen peroxide in water are approximately equal ($D_c = 2 \times 10^{-9} \, m^2 \, s^{-1}$)[48,49], therefore $\nabla \phi = \nabla[O_2] = -\frac{1}{2}\nabla[H_2O_2]$. Since only the concentration gradient matters for osmosis and phoresis, for simplicity we consider the sole quantity $\phi$. Consequently, the surface flow mobilities $\mu$ introduced in this study are associated with the gradients resulting from the hydrogen peroxide decomposition, rather than from a specific chemical compound. A second particle with mobility $\mu_p$ ($\mu_a$) will experience a slip velocity on its surface, $\boldsymbol{u}_s = \mu_p(\mu_a)\nabla_\parallel \phi$, that leads to a net diffusiophoretic velocity $\boldsymbol{V}_i$ ($\boldsymbol{v}_\alpha$). Here $\nabla_\parallel$ indicates the derivative tangential to particle surface.

As derived in "Materials and Methods" section, the speed of approach $\Delta v_r$ between an active and a passive particle at a relative distance $\Delta r$ reads:

$$\Delta v_r = v_\alpha + V = v_0 \left[ \bar{\mu} \left(\frac{\sigma_a}{\Delta r}\right)^2 + \frac{1}{4} \left(\frac{\sigma_p}{\sigma_a}\right)^3 \left(\frac{\sigma_a}{\Delta r}\right)^5 \right], \tag{3}$$

where $\bar{\mu} = \mu_p/\mu_a$ is the mobility ratio.

To determine the simulation parameters, we conducted several experiments where we measured the approach distance $\Delta r$ between an isolated pair of active and passive particles, Fig. 2A. We then calculated $\Delta v_r$ and used Eq. (3) to fit the experimental data. From the fit we find that the second term of the rhs in Eq. (3) is negligibly small and we extract a characteristic diffusiophoretic prefactor $v_0\bar{\mu} = 11.6 \pm 0.4 \, \mu m$ $s^{-1}$, Fig. 2B. The corresponding heat map of such a field is displayed in the inset of Fig. 2B, and was measured while keeping the orientation of the hematite fixed with a constant magnetic field. The iso-velocity lines are slightly elliptic, instead of circular as expected for an isotropic system. However, we find that this anisotropy is rather weak, and does not affect the raft dynamics. As shown in Supplementary Fig. 2, the probability distribution function for the angle $\tilde{\alpha}$, formed between the short axis of the hematite particle and the velocity vector $v_c$ of the raft, is flat within the error bars.

The simulations explain several experimental features: the growth of the raft size as $t^{1/3}$, the decrease of the raft velocity with the cluster area as shown in Fig. 2C, and the emergence of self-propulsion. This aspect is illustrated by Fig. 2D, which displays the average translation mean square displacement $MSD(\tau) \equiv \langle (\boldsymbol{r}(t) - \boldsymbol{r}(t+\tau))^2 \rangle \sim \tau^\delta$, with $\tau$ being the lag time and $\langle ... \rangle$ a time average. The MSD computed from experimental and simulation data shows both diffusive ($\delta = 1$) dynamics at short timescales, followed by a super diffusive ($\delta > 1$) dynamics, very close to a ballistic one ($\delta = 2$). However, the non overlapping MSD curves in Fig. 2D indicate that the simulations do not fully capture all the experimental features of the raft dynamics. Additionally, we measure the persistence length $l_p$ of the trajectory, i.e. the characteristic length over which the raft velocity orientation decorrelates. We calculate this quantity using the expression: $\langle \cos(\theta_v(d+\Delta l) - \theta_v(d) \rangle_d \propto \exp(-\Delta l/l_p)$ where $d$ is the distance traveled by the cluster and $\theta_v$ the orientation of the velocity vector. From the experiments, we measure $l_p \simeq 20 \, \mu m$ which is significantly larger than the one obtained from the numerical simulations, $l_p \simeq 2.5 \, \mu m$. Note that for an individual hematite particle we find a persistence length $l_p = 1.8 \, \mu m$. As we discuss below, this discrepancy arises from the opposite self-propulsion direction observed in experiments and in simulations for large clusters.

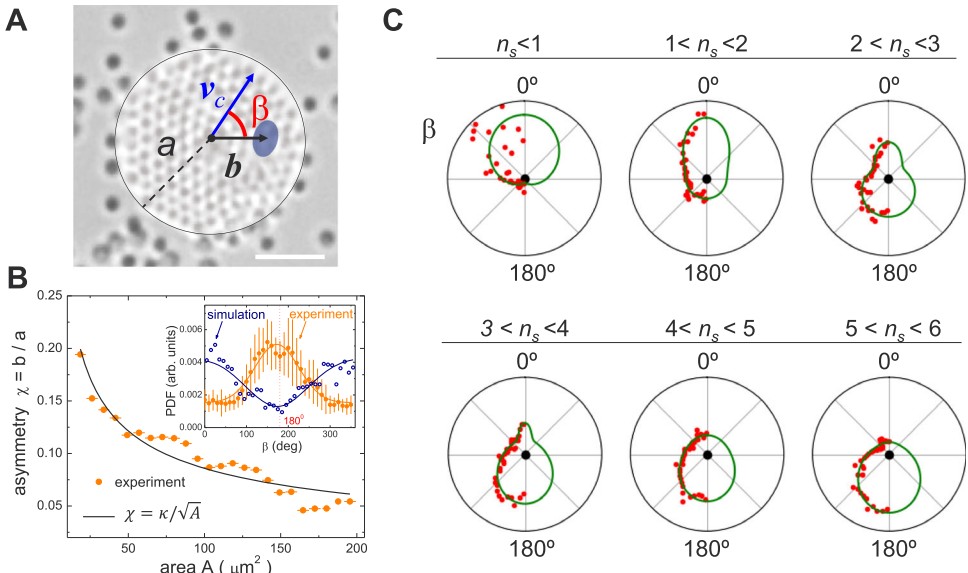

**Fig. 3 | Raft asymmetry. A** Schematic overlayed to a microscope image indicating the asymmetry vector $\boldsymbol{b}$ and the angle $\beta$ between $\boldsymbol{b}$ and the cluster velocity $\boldsymbol{v}_c$. The scale bar is 5 µm, however the location of the hematite in the scheme has been exaggerated (not on scale). **B** Experimentally measured asymmetry parameter $\chi = b/a$ versus cluster area $A$, being $a$ the cluster radius. The continuous line is an inverse square root law from which we extract the prefactor $\kappa = 0.858 \pm 0.019$ µm. Top left inset shows the distribution of angles $\beta$ between $v_c$ and the vector $\boldsymbol{b}$ pointing from the cluster center to the hematite particle from experiments (filled symbols) and simulations (open circles). In both cases the continuous lines are wrapped normal distributions and the error bars results from the statistical average of different measurements. **C** Polar plot showing the probability distribution from the experimentally measured angle $\beta$ (scattered red disks) for different clusters characterized by different number of shells $n_s$, where each shell is made of passive particles encircling the active one. The continuous green line is a non-linear regression of the data assuming wrapped normal distributions with one or two peaks. The different graphs have been obtained from a total of 30 separate experiments.

## Cluster asymmetry and self propulsion direction

To better understand the origin of the raft propulsion, we have analyzed the position of the hematite source within the cluster. During the growth process and at steady state, we find that the hematite is not at the geometric center of the cluster, but shifted from it by a small distance $b$, Fig. 3A. Furthermore, we find that the asymmetry parameter of the cluster, defined as $\chi = b/a$, decreases with the raft area $A$, where $a$ is the radius of the cluster. This dependency can be described by a power law, $\chi = \kappa A^{-1/2}$, Fig. 3B, which indicates that the variations of $b$ are rather small. Indeed, from the extracted prefactor $\kappa$, and taking into account that the radius of the cluster is $a = \sqrt{A/\pi}$, we deduce that $b = \kappa/\sqrt{\pi} = 0.48 \simeq \sigma_p/2$. In other words, the shift between the hematite and the cluster center is of the order of the radius of the passive particle, consistent with the growth of the cluster layer by layer.

The analysis of the distribution of the angles $\beta$ between the cluster velocity $v_c$ and the asymmetry vector $\boldsymbol{b}$ provides further insight into the propulsion direction. As depicted in the small inset of Fig. 3B, the overall distribution appears Gaussian (orange data and line) and centered around $\beta = 180°$, indicating that the raft propels with the active particle at the rear. However, in the numerical simulations the clusters tend to propel with the active particle at the front, as shown by the blue line and corresponding data in the same image, see also Supplementary Movie 4.

The position of the hematite in the raft controls its persistence length. This feature arises from the unfixed and evolving boundary that characterizes the self-propelled cluster of particles. Qualitatively, when a colloidal raft navigates through a crowded environment of passive particles, these particles tend to accumulate at the front. Consequently, a cluster propelled with the hematite shifted towards the front would have to regularly change its direction of motion to sustain this configuration, as reported in the simulations. On the other hand, if a cluster moves with the hematite shifted towards its rear, the colloids accumulate at its front, preserving the asymmetry and direction of motion, as observed in the experiments. These two situations result in a system with a relatively low and high persistence length, respectively. To validate this hypothesis, we have implemented a specific simulation where we have inverted the average velocity of the center of mass of the raft at each time step, while keeping the active-passive interactions outside the cluster in agreement with Eq. (3). As shown in Supplementary Movie 5, we find that the cluster moves with the hematite at the rear, and observe a longer persistence length, close to the experimental observations.

When examining the distribution of the angle $\beta$ for a given number of colloidal shells, $n_s$, we find that during the clustering process, the colloidal raft switches from a motion with the hematite at the front to rear. As shown in Fig. 3C, for the smallest clusters, up to two shells $n_s = 2$, the probability distribution of the angle $\beta$ exhibits a peak around 0°. This observation is consistent with the simulation predictions, and corroborates previous experiments conducted on clusters with fewer than one shell of passive particles[44]. For larger rafts, down to $n_s = 4$, the peak shifts to 180°. Due to the slow growing process, this orientation dominates the global dynamics described previously. For the intermediate sizes, $n_s = 2 - 4$, the distribution of angle $\beta$ exhibits coexisting peaks at opposite orientations, showing that the reversal in the direction of motion occurs within this size range.

The direction of the raft motion and the clustering effect cannot be explained by considering only a diffusiophoretic mechanism. Indeed both effects require a surface mobility, $\mu_p$, with opposite signs. The observation of clustering implies that $\mu_p < 0$, since otherwise the hematite would repel the silica colloids rather than attract them. On the other hand, assuming a purely phoretic phenomenon would imply that the cluster moves with the hematite at the front, as observed in the simulations. This effect is consistent with the experimental observations for small clusters ($n_s = 2$), but not for the larger ones. Moreover, assuming a purely diffusiophoretic phenomenon, a cluster moving with the colloids at the rear would require $\mu_p > 0$ (as we show later), which contradicts the observed clustering.

The discrepancy between the numerical and experimental results arises from the assumption of a purely diffusiophoretic system neglecting hydrodynamic interactions. The simulations do not account for the presence of the near wall, and the competition between diffusiophoresis and diffusioosmosis. Indeed, the importance of the proximity of the wall can be demonstrated by changing the nature of the substrate. In a separate set of experiments, we have repeated the assembly process above a polystyrene (PS) petri-dish. As shown in Supplementary Fig. 3, we observe a decrease in the cluster area compared to the glass. Moreover, the assembled raft did not exhibit any self-propulsion behavior, indicating that the modification of the osmotic effect due to the PS substrate also influences the raft's mobility. Specifically, a radial osmotic flow around the hematite and along the substrate develops, in opposition to clustering, promoting a vertical flow pressing the clusters against the wall due to incompressibility. In the case of PS, this force is strong enough to prevent the clusters from moving.

For the glass substrate, the change in the direction of motion of the raft can be interpreted as a switch from a self-propulsion process dominated by diffusiophoresis to diffusioosmosis. For the small clusters at the early stage, diffusiophoresis dominates, leading to a motion direction consistent with the simulation prediction. However, the clustering process modifies the aspect ratio of the colloidal raft. The emerging configuration, two mostly flat surfaces facing each other at a very short distance, favors the viscous interaction between the raft and the substrate. Thus, the clustering increases the viscous interaction with the substrate, up to the point where the substrate diffusioosmotic flows surpass the diffusiophoresis and determine the direction of motion.

## Theoretical model

The competition between diffusiophoresis and diffusioosmosis is the key element to understand the raft propulsion. Following the classical interpretation, one can consider that both phenomena arise from an osmotic effect[50]. More precisely, the local diffusioosmotic flows occurring on mobile surfaces lead to a slip velocity causing movement along the opposite direction[37]. Conversely, an external flow, as the osmotic flow occurring on a nearby substrate, tends to drag the object along the same direction. Then, if both surfaces generate a local osmotic flow along the same direction, their viscous interactions become opposite. Here we propose a more quantitative description illustrating the competition which occurs for the large rafts. To include the effect of hydrodynamics and the proximity of the wall, we approximate the colloidal raft by a disk of diameter $2a$ and the shifted hematite by a "semi-punctual" source, where the concentration field $\phi$ is similar to a punctual source except along the surface of the source, where $\phi$ is constant. We orient the system such that the unit vector $\mathbf{e}_z$ is diametrically opposed to the vector $\mathbf{b}$ linking the cluster center to the source. The negative or positive sign of the cluster velocity $v_c$ indicates whether a disk moving with the source at the front or the rear, respectively. We assume that the catalyzed product is released at the rate $J$, and diffuses in the bulk with a diffusion coefficient $D_c$. We consider two parallel surfaces, the disk ($p$) and the substrate ($S$), separated by $h$. If we assume that $h$ is roughly the height of the hematite ($h_a \simeq 0.3\,\mu m$), given the size of the cluster, we have $h/a = 0.07 \ll 1$.

To describe the disk dynamics we introduce two dimensionless numbers: the Péclet $\mathrm{Pe}_c = \frac{v_c a}{D_c}$ which compares the advective and diffusive transport, and the Damköhler number $\mathrm{Da} = \frac{\mu_p J}{4\pi a D_c^2}$ which relates the reaction rate to the diffusive mass transport rate. Experimentally, $\mathrm{Pe}_c \simeq 10^{-4} \ll 1$ thus, the solute transport is dominated by diffusion, and the source motion can be disregarded. At a distance $r$ from the source the chemical gradient can be expressed as $\nabla \phi = -J/(4\pi D_c r^2)\mathbf{e}_r$.

The concentration gradient generates a slip osmotic flow $\mathbf{u}_S = \mu \nabla_\parallel \phi$, along the surfaces of the disk $p$ and the substrate $S$, such that $\mathbf{u}|_p = v_c \mathbf{e}_z + \mu_p \nabla_\parallel \phi$, and $\mathbf{u}|_S = \mu_S \nabla_\parallel \phi$. The disk motion is force-free, hence $\mathbf{F}_v + \mathbf{F}_p + \mathbf{F}_S = 0$, where $\mathbf{F}_v$ is the damping force due to the motion of the disk, $\mathbf{F}_p$ is the phoretic force associated with the slip velocity on the disk's surface, and $\mathbf{F}_S$ the osmotic contribution coming from the slip velocity on the wall.

We determine the expression of the phoretic and osmotic forces using the Lorentz reciprocal theorem[51],

$$F_{p/S} = \int_{p/S} \frac{\mu_{p/S} \nabla_\parallel \phi}{v_c} \left(\mathbf{n} \cdot \hat{\mathbf{e}}\right) dS, \qquad (4)$$

where $\mathbf{n} \cdot \hat{\mathbf{e}}$ is the local viscous stress of the dual system counterpart, with $\mathbf{n}$ being the normal to the surface, and $\hat{\mathbf{e}}$ the viscous stress tensor. In this case, the dual counterpart of the problem involves an identical rigid particle as the model system, but without the osmotic flow on the surfaces. This is a disk sliding at velocity $v_c$ over the substrate. The expressions for the phoretic force $F_p$ and the osmotic force $F_S$ (Eq. (4)) correspond to the integration over surfaces—respectively the particle or substrate surfaces – of the dimensionless slip diffusioosmotic velocity, weighted by the local viscous stress of the dual problem. Considering the distribution of the viscous stresses, we find that the dominant contribution of both forces arises from the surfaces involved in the shear flow occurring between the disk bottom surface and the substrate in the dual problem. The details of all terms employed and the extended model are given in the Methods section.

By solving the force-free dynamic equation, we arrive at:

$$\mathrm{Pe}_c \simeq 2\,\mathrm{Da}(1 - \mu_S/\mu_p)\chi + O(\chi^2), \qquad (5)$$

and, accordingly, the velocity of the disk at the first order in $\chi$ is given by

$$v_c \propto (\mu_p - \mu_S)\frac{\chi}{A}. \qquad (6)$$

Note that if we remove the osmotic flow along the substrate, the term $\mu_S$ disappears from Eq. (6), and $v_c \propto \mu_p \frac{\chi}{A}$.

The difference between the osmotic mobilities $\mu_p - \mu_S$ in Eq. (6) marks the competition between diffusiophoresis and substrate diffusioosmosis, as it dictates the sign of $v_c$, i.e., the direction of motion of the raft. Since both the passive colloid and the substrate are made of silica, we assume that $\mu_S$ is comparable to $\mu_p$, and have the same negative sign. Thus, from the model and the observed direction of motion of the large cluster with the hematite at the rear, we deduce that $\mu_S/\mu_p > 1$. As a consequence, for the large cluster, diffusiophoresis acts against the motion, while the osmotic flow on the substrate induces the cluster propulsion. Moreover, our approach provides also the dependence of the disk velocity on $\frac{\chi}{A}$ that agrees well with the experimental observations, as shown in Fig. 4.

A characteristic feature of our system is the presence of a symmetric competition between osmotic and phoretic effects. This symmetry arises directly from the short distance $h$ relative to the cluster radius $a$. First, we can neglect the slight variation in concentration distribution between the disk bottom and the surface of the substrate, resulting in identical slip velocities modulo the mobility factor. Second and more importantly, in the dual problem, this thin-walled geometry induces a shear flow between the disk and the substrate, resulting in symmetric but opposite viscous stresses along the two surfaces. Since the distribution of the viscous stress of the dual surface weights the slip flow contribution in the expression of the osmotic/phoretic force, this feature is the origin

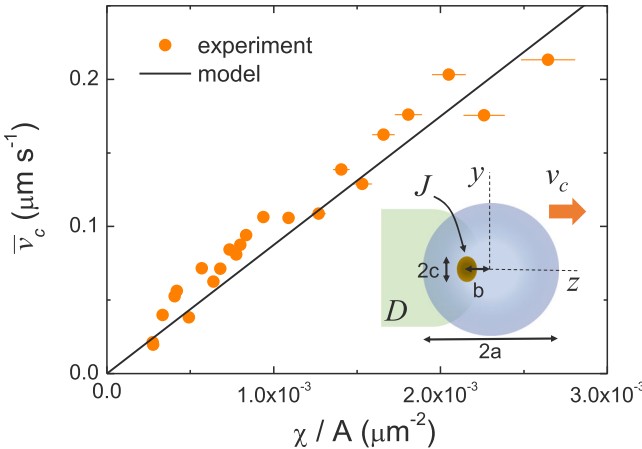

**Fig. 4 | Mean raft speed.** Experimental data of the mean cluster velocity $\bar{v}_c$ versus ratio $\chi/A$, being $\chi = b/a$. Scattered disks are experimental data while the continuous line is a fit from the theoretical model, Eq. (6) in the text. Inset illustrates a schematic of the model: the cluster is considered as a thin disk of radius $a$ with an active source of size $\sigma_a$ at a distance $b$ from the center. $J$ and $D$ denote respectively the release rate of the source and the solvent diffusion rate. The error bars results from the statistical average of different measurements.

of the symmetric competition between osmotic and phoretic forces for large clusters.

If we consider a scenario where the condition $h/a \ll 1$ is not satisfied, such as when the disk size becomes comparable to the distance $h$, the simple shear flow might disappear in the dual problem. Consequently, the viscous stress on the bottom of the disk becomes larger than that on the substrate, breaking the symmetry of the competition in favor of the phoretic force. This explains the observed transition in the dominant force between osmosis and phoresis for small to large clusters.

## Discussion

We have investigated the rich dynamics of active colloidal rafts composed of a central hematite particle and several shells of passive colloids. We have shown that this system displays a clustering phenomenon due to diffusiophoresis, and collective self-propulsion that result from the interplay between diffusiophoresis and diffusioosmosis on a nearby substrate. While the first mechanism dominates for small clusters, hydrodynamics become important and prevail for a number of shells $n_s > 3$. Indeed, simulations based solely on diffusiophoresis accurately describe the clustering kinetics but cannot explain the direction of motion and persistence length for large clusters. Moreover, a purely diffusiophoretic mechanism cannot account for both clustering and the direction of the cluster motion, since both effects would imply surface mobility on the silica colloids characterized by opposite signs.

Our model resolves this discrepancy by considering the cluster asymmetry and the substrate diffusioosmotic flow. Thus, we have shown that there is a competition between the diffusiophoresis and osmosis, and the crucial role of the substrate diffusioosmotic flow on the raft dynamics. In line with these results, previous works in the field have also shown the importance of considering the osmotic flow generated by an active particle near a wall[13,52]. The theoretical approach, based on the Lorentz reciprocal theorem, could be extended to many other catalytic active systems close to a substrate, considering the proper boundary conditions. In our experiments, we approximate the raft as a disk enabling to derive an analytical expression that captures the underlying physics of this complex, yet rich hybrid active passive system.

## Methods

### Experimental details

The active particles are hematite prolate ellipsoids synthesized following the "gel-sol" technique introduced in ref. 53. While maintaining the stirring, an iron chloride hexahydrate solution (54.00 g in 100 mL of water, Sigma-Aldrich 31232-M) is gradually added to a sodium hydroxide solution (19.48 g in 90 mL of water; Sigma-Aldrich S5881), followed 5 min after by a potassium sulfate solution (0.29 g in 10 mL of water; Sigma-Aldrich P0772). After being agitated for 5 more minutes, the mixture is hermetically sealed in a 1L bottle and left to age at 100 °C for 8 days. Afterwards, the reaction is stopped by filling the recipient with water and leaving it to cool down in a fridge. The hematite particles are concentrated and washed through multiple cycles involving centrifugation and a re-dilution in clean deionized water.

The experimental system consists of a colloidal suspension made of synthesized hematite particles mixed with silica spheres (diameter 1 μm; Sigma-Aldrich) dispersed into an aqueous solution of hydrogen peroxide (3.6% w/v; Fisher BP2633). The quantity of active particles is much lower than passive ones, the ratio is below 1 active for 2000 passives. The solution is made basic (pH ~ 9.2) by adding TMAH (Sigma-Aldrich 328251), to see the attraction phenomenon, and treated for 5 minutes with an ultrasound bath to break up the hematite chains. Right after, we introduce by capillarity the colloidal mixture into a rectangular glass micro-tube (inner dimension 2 × 0.1 mm; CMC Scientific) immediately sealed with wax at the atmospheric pressure. After few minutes, the colloidal particles sediment close to the bottom of the glass micro-tube, such that they form a quasi-two-dimensional system with a distance of 100 μm from the top wall. The high pH ensures the stability of the dispersion and prevents the colloids from sticking to the glass substrate. Contrary to ref. 54, no surfactant is added to the solution since we have observed that SDS lowers the attraction phenomena.

To record the colloid raft behavior, we use an upright optical microscope (Eclipse Ni; Nikon) equipped with a charge-coupled device camera (12 up to 50 frame per second; Basler Scout scA640-74f) and an epifluorescent tower. For the illumination, the light is provided by a commercial mercury fiber illumination system (C-HGFI Intensilight; Nikon) filtered with a band-pass filter (Nikon B-2A filter). The output after the objective (Nikon MRH01902) is a blue light (wavelength $\lambda = 450$–$490$ nm) with an intensity going from 4 up to 125 mWcm$^{-2}$. An image of the experimental setup is given in Supplementary Image 1.

We have independently checked the effect of the illumination power on the eventual variation of the temperature within the experimental system or the presence of drift due to light absorption. As shown in Supplementary Fig. 5, we have tracked the position of silica spheres with 2 micron diameters for different applied powers, and measured the MSDs from which we extract the corresponding diffusion coefficients. We find that, in all cases, the MSDs are diffusive and isotropic along the two orthogonal directions in the particle plane $(x, y)$. Additionally, we provide in Supplementary Note 2 more experimental details on the methods employed for measuring the surface area $A$ of the colloidal raft, the raft velocity $v_c$, the relative velocity between two colloids $\Delta v_r$, and the geometric quantities $a$, $b$ and the angle $\beta$.

### Numerical simulations

We start with the details on how we derive Eq. (3) in the main text. We consider that the active and passive particles of diameters $\sigma_a$ and $\sigma_p$, respectively, are immersed in a concentration field $\phi(r)$. The concentration field obeys the Laplace equation $\Delta c(r, \theta) = 0$, where Neumann boundary conditions are applied such that $D_c \partial_r c(r, \theta)|_\phi \propto \alpha_r$ on each particle's surface. In the previous expression $\alpha_r$ is the production/consumption rate of the chemical, and $D_c$ its diffusion coefficient. For passive particles $\alpha_r = 0$, as they do not consume or produce any chemical. We assume, for simplicity, that fuel depletion is negligible.

An active particle, $i$, located at the origin and in the presence of a second particle, $j$, at a distance $d_{ij} = |\mathbf{r}_i - \mathbf{r}_j|$, with the center-to-center direction parallel to $\hat{z}$, creates a concentration field

$$c_i(r, d, \theta) = \frac{\alpha_{r,i}\sigma_i^2}{4D_c}\frac{1}{r} + \phi_{i,j}(r, d_{ij}, \theta). \tag{7}$$

The first term on the right-hand side gives the production of chemical, while the second describes the disturbance of the chemical concentration produced by the $j$-th particle and guarantees that the boundary condition is satisfied on the $i$-th particle. The disturbance term $\phi_{ij}$ can be expanded as a multipolar series with the axis of symmetry along $\hat{z}$.

The first contribution on the particle $i$ created by a particle $j$, $\phi_{i,j}^{(1)}(r, d_{ij}, \theta)$, corresponds to a dipole and depends on the activity $\alpha_{r,j}$ of the $j$-th particle, and the distance between the pair

$$\phi_{i,j}^{(1)}(r, d, \theta) = -\frac{1}{2}\left(\frac{\sigma_i}{2}\right)^3 \frac{1}{d_{ij}^2}\frac{\alpha_{r,j}\sigma_j^2}{4D_c}\frac{\cos\theta}{r^2}, \tag{8}$$

which for an active particle producing a concentration field in the presence of a passive corresponds to $\phi_{a,p}^1 = 0$. The second dipolar contribution on the particle $i$ appears only on the active particles. The monopolar field generated by $i$ is reflected on particle $j$ and gives, on $i$ the dipolar term, $\phi_{i,j}^{(2)}$. As expected it depends on $\alpha_{r,i}$ as:

$$\phi_{i,j}^{(2)}(r, d, \theta) = -\frac{1}{4}\left(\frac{\sigma_i}{2}\right)^3\left(\frac{\sigma_j}{2}\right)^3\frac{1}{d_{ij}^5}\frac{\alpha_{r,i}2\sigma_i^2}{4D_c}\frac{\cos\theta}{r^2}. \tag{9}$$

Note that this last contribution cancels for $i = p$, leading to the non-reciprocity of the interactions between active and passive particles. The gradient of the chemical concentration on the surface of a sphere generates a tangential diffusiophoretic velocity, $\mathbf{u} = \mu_d\nabla_\parallel c(\mathbf{r})$, of the fluid at the particle interface. Here, $\nabla_\parallel$ is the gradient calculated at the surface of the sphere, which is tangential to the sphere at every point of the surface $\nabla_\parallel = (\mathbf{I} - \hat{n}\hat{n}) \cdot \nabla$, where $\mathbf{I}$ is the unity matrix and $\hat{n}$ is the vector normal to the surface.

The particle velocity can be obtained from the diffusiophoretic velocity by imposing the momentum conservation

$$\mathbf{V} = -\frac{1}{\pi\sigma^2}\int d\Omega \mathbf{u}(r, \theta) = (\pi\sigma^2)^{-1}\mu_d \int d\Omega \nabla_\parallel c(r, \theta). \tag{10}$$

The integration of $\nabla_\parallel\phi$ for a multipolar expansion of the form $c(\theta, r) = \sum_l B_l P_l(\cos\theta)r^{-(l+1)}$ on a spherical shell of diameter $\sigma$ results in a velocity in $\hat{z}$, the symmetry axis of the system,

$$\mathbf{V} = \frac{2}{3}\mu_d\left(\frac{2}{\sigma}\right)^3 B_1\hat{z}. \tag{11}$$

Hence, the diffusiophoretic velocity of the particle depends only on the $l = 1$ contribution of the multipolar expansion of the chemical field around the particle center.

Introducing Eqs. (8)–(9) into Eq. (11), we recover the relative velocities (Eq. (3) of the main text), with a characteristic velocity $v_0 = \alpha_r\sigma_a^2\mu_a/(12D_c)$. We have taken into account that active and passive particles have different diffusiophoretic mobilities, $\mu_a, \mu_p$, and have introduced their ratio, $\bar{\mu} = \mu_p/\mu_a$.

Thus, the passive particle velocity induced by an active one is

$$\mathbf{V}_i = \sum_{j\neq i} v_0\bar{\mu}\left(\frac{\sigma_a}{d_{ij}}\right)^2\hat{\mathbf{r}}_{ij}, \tag{12}$$

while the velocity of an active particle induced by the passive particle due to the diffusiophoresis reads:

$$\mathbf{v}_i = \sum_{j\neq i} \frac{V_0}{4}\left(\frac{\sigma_p}{\sigma_a}\right)^3\left(\frac{\sigma_a}{d_{ij}}\right)^5\hat{\mathbf{r}}_{ij}. \tag{13}$$

Following the notation in the main text, we consider $N$ passive particle of diameter $\sigma_p$ and mobility $\mu_p$ in the presence of two active particles forming a dumbbell. The active particle consists of two spheres each with mobility $\mu_a$, diameter $\sigma_a$, and separated by spring of rest length $l_0$. To reduce the number of parameters, we consider the dimensionless length $r = r'\sigma_p$, and the dimensionless time $t = \tau_c t'$, using the characteristic time $\tau_c = \sigma_p/v_0\mu$. Thus, we convert the simulations units to experimental units by means of the characteristic length $\sigma_p = 1\mu m$ and the characteristic time $\tau_c = 0.086s$, which is directly measured from the pre-factor $v_0\bar{\mu}$. The equations of motion can be rewritten in terms of these dimensionless variables as

$$\dot{\mathbf{r}}_\alpha = \sum_j \frac{\hat{\mathbf{r}}_{\alpha j}}{\bar{\mu}}\left(\frac{\sigma_p^3\sigma_a^2}{r_{\alpha j}^5}\right) + \frac{\mu_a(F^h\hat{\mathbf{n}}_\alpha + \mathbf{F}_\alpha^c)}{v_0\bar{\mu}} + \sqrt{\frac{2\tau_c}{Pe_a}}\hat{\boldsymbol{\xi}}_\alpha, \tag{14}$$

$$\dot{\mathbf{R}}_i = \sum_{j\neq i}\left(\frac{\sigma_a}{r_{ij}}\right)^2\hat{\mathbf{r}}_{ij} + \frac{\mu_p}{v_0\bar{\mu}}\mathbf{F}_i^c + \sqrt{\frac{2\tau_c}{Pe_p}}\hat{\boldsymbol{\xi}}_i, \tag{15}$$

where $Pe_{a,p} = \sigma_p v_0\bar{\mu}/D_{a,p}$ is the Péclet number, and subindices $a$ and $p$ distinguish between the active and passive Péclet, since they correspond to particles with different diffusion coefficients. The noise is included in the dimensionless stochastic variable $\hat{\boldsymbol{\xi}}$, which follows a Gaussian distribution $\langle\hat{\boldsymbol{\xi}}\rangle = 0$, and has unit variance. We briefly discuss the rest of the forces separately and highlight how the dimensionless parameters are chosen. First, we note that the subindex $\alpha = 1, 2$ refers to each bead constituting the dumbbell joined by the harmonic spring interaction $F^h = -k|\mathbf{r}_1 - \mathbf{r}_2|$. The dimensionless form of the harmonic spring term, corresponding to the second term in the rhs in Eq. (14) is:

$$\mathbf{F}_h' = \frac{\mu_a k\sigma_p}{V_0\mu}\mathbf{n}_\alpha|\mathbf{r}_1' - \mathbf{r}_2'|. \tag{16}$$

We choose the dimensionless spring parameter, $\mu_a k\sigma_p/V_0\mu = 50$ so that the spring joining the two beads is rigid and almost incompressible, resulting in a rigid active particle. $F_c$ and $F_c^\alpha$ follow a standard Weeks-Chandler-Andersen (WCA) potential. The dimensionless expression for this force is

$$\mathbf{F}_{ij}^{c'} = \frac{24\mu_a\varepsilon_{\delta\beta}}{v_0\bar{\mu}\sigma_{\delta\beta}}\left[2\left(\frac{\sigma_{\delta\beta}}{r_{ij}'\sigma_p}\right)^{13} - \left(\frac{\sigma_{\delta\beta}}{r_{ij}'\sigma_p}\right)^7\right]\hat{\mathbf{r}}_{ij}, \tag{17}$$

being $\sigma_{\delta\beta} = 0.5(\sigma_\delta + \sigma_\beta)$, where $\delta, \beta = \{a, p\}$ stands for any combination pair of passive and active particles. Following the standard WCA force, we impose a cutoff at the minimum $2^{1/6}\sigma$, such that the force is purely repulsive. To simulate the excluded volume interactions, we choose $\mu_a\varepsilon_{\alpha\beta}/v_0\bar{\mu} = 5$, for any combination of active and passive interactions. The system dynamics are dominated by $Pe_{a,p}$ which are the Péclet number of the active and passive particles resp., and by the mobility ratio $\bar{\mu}$. Note that in dimensionless units, from Eq. (14) a large value of $\bar{\mu}$ implies a small contribution of the velocity induced by passive particles on the active one. This is indeed the situation observed in the experiments. In the simulations, we give to $\bar{\mu}$ a value which is compatible with that obtained from the fit in Fig. 2(B), $\bar{\mu} \sim 10$. Around this value, we performed additional simulations, to determine whether small deviations may affect the cluster formation. We observed that

the MSD and the cluster growth of the active raft did not change quantitatively. Finally, to determine the Péclet numbers, we use the diffusion coefficients extracted from the experiments ($D_p$, $D_a$). Thus, we use the values $Pe_a$=53.8 and $Pe_p$=60, such that that they fulfill the relationship $Pe_a/Pe_p=D_p/D_a$. We performed 24 simulations including $N_p = 700$ passive particles and $N_a = 1$ active dumbbell, each simulation performed with different random initial positions of all particles. We use the same area fraction as in the experiments, $A_\varphi = 0.06$. The total simulation time for each run is $t_{tot} = 60000\tau_c = 5160$ s, snapshots for analysis are taken every $t_f = 1.4\tau_c = 0.12$ s, and the time step is $dt = 5 \cdot 10^{-4}\tau_c = 4.35 \cdot 10^{-5}$ s. All the parameters used for the numerical simulations are provided in Supplementary Table 1.

The cluster radius $a$ can be estimated at the steady state, when the diffusiophoretic interaction becomes comparable to the averaged forces due to thermal noise. Comparing the diffusiophoretic term from Eq. (14) and the diffusion one, Eq. (15) we get

$$a \sim \frac{\sigma_a}{\sqrt{\langle \hat{\boldsymbol{\xi}}^2 \rangle}} \left(\frac{Pe_p}{2}\right)^{1/4}. \tag{18}$$

Using the experimental value of $Pe_p$=60, and approximating the hematite particle as a spherical one with an equivalent radius of $\sigma_a \simeq 1.55 \,\mu m$, we obtain $a \simeq 9\,\mu m$, in agreement with numerical simulations. Thus, the growth rate of the cluster is compatible with an attraction $1/r^2$. Note that the diffusion coefficient $D_p$ controls the size of the raft. This result has been confirmed in the simulations by varying $Pe_p$ inside the range compatible with the experimental measurements. Supplementary Fig. 6(a) shows how the cluster grows its area compared to an expression of the type $t^{1/3}$, similar to the experiments. In Supplementary Fig. 6(b) the cluster has a diffusive behavior at small times and moves ballistically at large ones, due to the non-reciprocal interactions. Supplementary Fig. 6(c) shows how the velocity decreases as a function of time. Supplementary Fig. 6(d) shows how the velocity follows approximately a linear behavior as a function of $\chi/A$, as suggested by the theoretical results.

### Details of the theoretical model

**Phoretic and osmotic expression with the Lorentz theorem.** In the context of phoretic/osmotic system, the main interest of the reciprocal theorem is to directly compute integral quantities such as the viscous force without solving the Stokes flows occurring in this type of system. The motion of an object in a concentration gradient is attributed to the osmotic flow $\boldsymbol{u}_S$ along its surface $S$,

$$\boldsymbol{u}_S = \mu \boldsymbol{\nabla}_\| \phi, \tag{19}$$

with $\phi$ a chemical concentration and $\mu$ the osmotic mobility of the slip flow.

For the most general case, we consider an object moving along a substrate at the velocity $v_c \boldsymbol{e}_z$ in a gradient of concentration $\boldsymbol{\nabla}\phi$. The concentration gradient generates a slip osmotic flow Eq. (19) along the object surface $p$ and the substrate $S$, such that:

$$\boldsymbol{u}|_p = v_c\,\boldsymbol{e}_x + \mu_p \boldsymbol{\nabla}_\| \phi, \qquad \boldsymbol{u}|_S = \mu_S \boldsymbol{\nabla}_\| \phi. \tag{20}$$

The use of the Lorentz reciprocal theorem requires the introduction of a dual system. The dual system shares at any moment the same boundaries, but no osmotic phenomenon occurs, neither on the object surface nor on the substrate. In this situation, we simply assume a no slip boundary condition:

$$\hat{\boldsymbol{u}}|_p = v_c\boldsymbol{e}_z, \qquad \hat{\boldsymbol{u}}|_S = \boldsymbol{O}. \tag{21}$$

To distinguish the quantities describing the dual problem we add a hat " $\hat{\,}$ " to them. According to the Lorentz reciprocal theorem[51], we have

$$\int_p (\boldsymbol{n} \cdot \boldsymbol{\epsilon}) \cdot \hat{\boldsymbol{u}}\,dS + \int_S (\boldsymbol{n} \cdot \boldsymbol{\epsilon}) \cdot \hat{\boldsymbol{u}}\,dS = \int_p (\boldsymbol{n} \cdot \hat{\boldsymbol{\epsilon}}) \cdot \boldsymbol{u}\,dS + \int_S (\boldsymbol{n} \cdot \hat{\boldsymbol{\epsilon}}) \cdot \boldsymbol{u}\,dS, \tag{22}$$

with $\boldsymbol{n}$ the surface normal oriented toward the fluid and $\boldsymbol{\epsilon}$ the stress tensor, such that $\boldsymbol{\epsilon} = -p\mathbb{1} + \eta(\boldsymbol{\nabla}\boldsymbol{u} + \boldsymbol{\nabla}\boldsymbol{u}^T)$, with $p$ the pressure and $\eta$ the fluid viscosity. Knowing the boundary condition for the dual system, the left-hand side simplifies into $v_c F_v$. By using the boundary condition (21)(20) in the integrals of the left-hand side, we obtain an expression for the total viscous force $F_v$ as a sum of three other forces,

$$\boldsymbol{F}_v = \hat{\boldsymbol{F}}_v + \boldsymbol{F}_p + \boldsymbol{F}_S, \tag{23}$$

$$\hat{\boldsymbol{F}}_v = \int_p \boldsymbol{n} \cdot \hat{\boldsymbol{\epsilon}}\,dS, \tag{24}$$

$$\boldsymbol{F}_p = \int_p \frac{\boldsymbol{n} \cdot \hat{\boldsymbol{\epsilon}}}{v_c} \cdot \left(\mu_p \boldsymbol{\nabla}_\| \phi\right) dS\,\boldsymbol{e}_z, \tag{25}$$

$$\boldsymbol{F}_S = \int_S \frac{\boldsymbol{n} \cdot \hat{\boldsymbol{\epsilon}}}{v_c} \cdot \left(\mu_S \boldsymbol{\nabla}_\| \phi\right) dS\,\boldsymbol{e}_z, \tag{26}$$

being $\boldsymbol{u}$ the flow field, $\boldsymbol{n}$ the surface normal oriented toward the fluid. $\hat{\boldsymbol{F}}_v$ is the viscous drag of the dual problem, similar to a damping force since $\hat{\boldsymbol{F}}_v \propto -v_c\boldsymbol{e}_z$, $\boldsymbol{F}_p$ is the viscous force due to the osmotic flow along the moving object, i.e. the phoretic force, and $\boldsymbol{F}_S$ is the viscous force due to the osmotic flow along the wall, i.e. the osmotic force.

For a system moving at a constant velocity, the force balance imposes that the total viscous force is null: $\boldsymbol{F}_v = \boldsymbol{O}$. It leads to the dynamic equation,

$$\boldsymbol{O} = \hat{\boldsymbol{F}}_v + \boldsymbol{F}_p + \boldsymbol{F}_S. \tag{27}$$

Note for a diffusiophoretic spherical object far from any wall, the tangent stress is constant along the surface[40] for the dual problem. We deduce the well-known result[50] that the phoretic velocity is the opposite of the average slip velocity (19) over the object surface $\boldsymbol{v}_c = -\langle \boldsymbol{u}_p \rangle_p = -\langle \mu_p \boldsymbol{\nabla}_\| \phi \rangle_p$.

**The Raft motion.** To model the cluster motion, we approximate the colloidal raft by a disk and the central hematite by an "semi-punctual" source with a radius $\sigma_a/2$, i.e the concentration field is the same as a punctual source except along the source surface where is constant. We assume that the catalyzed product is released at the total rate $J = \alpha_r\pi\sigma_a^2$, and diffuses in the bulk at the diffusion rate $D_c$. The shift from the disk center is $b$, while the disk radius is $a$ with an area $A$. We orient the system such that the unit vector $\boldsymbol{e}_z$ is diametrically opposed to the vector $\boldsymbol{b}$ linking the cluster center to the source. Thus, the disk velocity $v_c\boldsymbol{e}_z$ can be positive or negative, which indicates respectively a self-propulsion with the source at the rear or at the front.

We introduce the Péclet number $Pe_c = \frac{v_c a}{D_c}$, the Damköhler number $Da = \frac{\mu_p J}{4\pi a D_c^2}$, the asymmetry number $\chi = \frac{b}{a}$ and also the dimensionless substrate osmotic mobility $\bar{\mu}_S = \frac{\mu_S}{\mu_p}$.

Experimentally $Pe_c \ll 1$, thus we assume that the motion of the source does not affect the chemical distribution $\phi$, and by extension, the concentration gradient. For $r > \sigma_a/2$,

$$\phi = c_0 + \frac{J}{2\pi D_c r}, \qquad \boldsymbol{\nabla}\phi = -\frac{J}{2\pi D_c}\frac{1}{r^2}\boldsymbol{e}_r, \tag{28}$$

with $r$ the distance from the source, and $c_0$ the concentration at the infinity. We neglect the terms in $O(h^2)$ due to the short distance $h$ between the source and the impermeable substrate.

Because of the electrostatic interaction, the cluster is sliding at a distance $h$ from the substrate, with $h$ small compared to the cluster size. Two boundary conditions might be studied for the substrate, either no-slip as we consider in the wall model, or implementing an osmotic flow as we consider in the osmotic model. From the point of view of the Lorentz reciprocal theorem, both models are very similar to treat, since they share the same dual problem. The only difference is the osmotic force $F_S$, which is null for the wall model.

**Surface stress in the dual problem.** We seek analytical expression for the viscous stress $\hat{\boldsymbol{\epsilon}}$. The dual problem is a disk sliding at the distance $h$ over a substrate. Even if the problem has already been solved analytically[55], the complexity of the expressions does not fit the objective of introducing analytical toy models with a simple view of the physics at play. We propose simple approximations for the viscous stress applying on the different surfaces.

**Disk upper side.** We assume that the disk upper side facing the infinite half-space of the substrate undergoes the same stress with or without the wall. Then, the tangential viscous stress on the disk surface is given by

$$\boldsymbol{n}.\hat{\boldsymbol{\epsilon}} = -\eta \frac{v_c}{a} \frac{C}{2\pi} \frac{1}{\sqrt{1 - r^{*2}}} \boldsymbol{e}_z, \tag{29}$$

with $r^*$ the distance from the disk center and $C = 16/3$. For an oblate the stress reaches its maximum intensity over the perimeter[51], for a disk it diverges over a circumference $r^* = 1$.

**Disk lower side and substrate right below the disk.** For the lower disk side and the substrate straight below, we assume the velocity profile is a solution of the Stokes flow in the presence of a solid boundary. Thus, $\hat{\boldsymbol{u}}(x, y, z) = Ay^2 + By\,\boldsymbol{e}_z$, with two boundary conditions: $\hat{\boldsymbol{u}}(x, y = h, z) = v_c\,\boldsymbol{e}_z$ and $u(x, y = 0, z) = 0$. At the disk border, we impose that the pressure is equal to $P_0$, therefore the only valid solution is a pure shear flow

$$\boldsymbol{u}(x, y, z) = v_c \frac{y}{h} \boldsymbol{e}_z, \qquad \boldsymbol{n} \cdot \hat{\boldsymbol{\epsilon}} = (\boldsymbol{n} \cdot \boldsymbol{e}_z) \frac{v_c \eta}{h} \boldsymbol{e}_z, \tag{30}$$

with $\boldsymbol{n}$ the normal to the surface considered.

**Remaining substrate.** For the remaining surface of the substrate, we assume that its contribution in the osmotic force $F_S$ is negligible compared to the contribution of the surface right below the disk, considering that $\boldsymbol{n} \cdot \hat{\boldsymbol{\epsilon}} \cdot \nabla_\| \phi \propto \frac{1}{r^4}$.

**Forces in the osmotic/phoretic problem.** Knowing the stress over the surface in the dual problem and the chemical concentration gradient (28), we determine the forces applied to the disk. The damping force is:

$$\hat{\boldsymbol{F}}_v = -\eta v_c a \left( C + \frac{\pi}{\bar{h}} \right) \boldsymbol{e}_z, \tag{31}$$

with $\bar{h} = h/a$. The drag formula proposed is computed from the rough stress estimation described above. Compared to the right variation, it becomes asymptotically accurate for $\bar{h} \ll 1$[56], when the shear stress on the bottom side dominates. The Phoretic force on the upper face is:

$$\boldsymbol{F}_p^{up} = \frac{C}{\pi} D_c \eta \, \mathrm{Da} \int_{\theta=-\pi}^{\theta=\pi} \cos\theta \int_{r=\bar{c}}^{r=\hat{a}} \frac{1}{\bar{r}\sqrt{1-\bar{r}^{*2}}} \, \mathrm{d}\bar{r} \, \mathrm{d}\theta \, \boldsymbol{e}_z, \tag{32}$$

with $\quad \hat{a} = \chi \cos\theta + \sqrt{1 - \chi^2 \sin^2\theta},$

$$\bar{r}^{*2} = \bar{r}^2 \left( 1 + \left( \frac{\chi}{\bar{r}} \right)^2 - 2 \frac{\chi}{\bar{r}} \cos\theta \right), \quad \text{and} \quad \bar{c} = \frac{c}{a}. \tag{33}$$

In the above expression, the surface integral is expressed in the cylindrical frame of reference centered on the source. Some quantities need to be rewritten in this frame, such as the radial position of the disk perimeter $\hat{a}(\theta, \chi)$ and the distance from the disk center $\bar{r}^*(\theta, \chi)$.

The Phoretic force on the lower face and osmotic force is:

$$\boldsymbol{F}_p^{down} = 2 D_c \eta \frac{\mathrm{Da}}{\bar{h}} \int_{\theta=-\pi}^{\theta=\pi} \cos\theta \int_{r=\bar{c}}^{r=\hat{a}} \frac{1}{\bar{r}} \, \mathrm{d}\bar{r} \, \mathrm{d}\theta \, \boldsymbol{e}_z, \tag{34}$$

$$\boldsymbol{F}_S = -2 D_c \eta \frac{\mathrm{Da}\,\bar{\mu}_S}{\bar{h}} \int_{\theta=-\pi}^{\theta=\pi} \cos\theta \int_{r=\bar{c}}^{r=\hat{a}} \frac{1}{\bar{r}} \, \mathrm{d}\bar{r} \, \mathrm{d}\theta \, \boldsymbol{e}_z. \tag{35}$$

Finally, by using the dynamic Eq. (27), we obtain an equation involving the dimensionless numbers,

$$\mathrm{Pe}_c = \frac{2\mathrm{Da}}{\bar{h}C + \pi} \int_{\theta=-\pi}^{\theta=\pi} \cos\theta$$

$$\int_{r=\bar{c}}^{r=\hat{a}} \frac{1}{\bar{r}} \left( \frac{\bar{h}C}{2\pi\sqrt{1-\bar{r}^{*2}}} - \bar{\mu}_S + 1 \right) \mathrm{d}\bar{r} \, \mathrm{d}\theta. \tag{36}$$

If we assume $\bar{h} \ll 1$ and $\chi \ll 1$, the contribution of the upper face becomes negligible. We reach the formula:

$$\mathrm{Pe}_c \simeq 2\,\mathrm{Da}(1 - \bar{\mu}_S)\chi + O(\chi^2), \qquad v_c \propto (\mu_p - \mu_S)\frac{\chi}{A}. \tag{37}$$

Note that if we remove the osmotic flow along the substrate, we have instead for the wall model

$$\mathrm{Pe}_c \simeq 2\mathrm{Da}\chi + O(\chi^2), \qquad v_c \propto \mu_p \frac{\chi}{A}. \tag{38}$$

## Data availability
Source data for all figures are provided with the paper as a Source Data file. The data that support the findings of this study are available from the corresponding author upon request.

## Code availability
The computer code is provided with the paper.

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

## Acknowledgements

This work has received funding from the European Research Council (ERC) under the European Union's Horizon 2020 Research and Innovation Program (grant agreement no. 811234). S. G. L. and I. P. acknowledge support from Ministerio de Ciencia, Innovación y Universidades (grant no. PID2021-126570NB-100 AEI/FEDER-EU) and from Generalitat de Catalunya under project 2021SGR-673. P. T. acknowledges support from the Ministerio de Ciencia e Innovació (Project No. PID2022-137713NBC22), the Agència de Gestió d'Ajuts Universitaris i de Recerca (Project No. 2021 SGR 00450). P. T. and I. P. acknowledge support from the Generalitat de Catalunya (ICREA Acadèmia).

## Author contributions

D.B. performed the experiments and developed the theoretical model. S.G. ran the numerical simulations. I.P. and P.T. supervised the work. All authors discussed the results and commented on the manuscript at all stages.

## Competing interests

The authors declare no competing interests.
