## [Peer Review File · Nature Communications]

REVIEWER COMMENTS

Reviewer #1 (Remarks to the Author):

The manuscript 'Clustering induces switching between phoretic and osmotic propulsion in active colloidal rafts' describes a combination of experiments and simulations of self-assembled and self-propelling 'rafts'. Each raft is composed of silica particles that self assemble around a catalyst particle under the exposure to blue light.

This work builds upon the authors previous results with studying self-assembled dimers (and larger assemblies) of the same passive and active particles (Reference 43). In this previous study the role of diffusiophoresis in the aggregation process was the main focus. In this new study, the authors focus on the self-propelling behaviour of the larger assemblies (more than 4 shells of passive particles) where the experiment showed an inversion of the mobility of the rafts and the persistence length of the raft motion. The persistence length observed for larger rafts seems larger than what could be explained with the previous model that only takes diffusiophoresis into account. The inversion of mobility refers to the fact that for some raft sizes the propulsion is with the catalyst particle at the rear of the raft while for other sizes the propulsion is with the catalyst at the front, with respect to the center of the raft. The authors then derive a new model that takes into account the diffusioosmotic flow induced at the substrate. The new model then recapitulates the observed mobility inversion. This study thus shines new light on the interpretation of an experiment where diffusiophoresis was considered while neglecting its counterpart, diffusioosmosis. This is certainly of value to the field of colloidal self-assembly and self-propelling colloids. And so I can recommend the manuscript for publication.

The manuscript is well structured and figures are of high quality, however the manuscript would clearly benefit from proofreading. Also, the authors assume a lot when exposing the simulations. I would suggest revising the way the model is presented to make it more accessible, eventually recapitulating key elements of previous papers, and adding numerical values.

Major:

It occurred to me quite late in the manuscript that the only chemical gradient considered in this system is the gradient of the reaction product oxygen. The gradient of reagents due to local depletion near the catalyst particle is neglected. This was first clear to me when reading ref 43. Even then, the argument for neglecting it is still unclear. Is that because $\text{grad } c / c_0$ is negligible? I think it needs to be clarified early in the ms.

How far does the concentration gradient extend around the catalytic particle? How is the length scale in figure 2B related to the maximum number of shells obtained for a given illumination power in figure 1D? Is the saturation of the raft size directly related to a gradient threshold reached at this distance from the catalytic particle? This is not clear.

Since only the concentration of oxygen matters, could the author give the diffusion coefficient of oxygen used, the rate J , the mobilities etc.? Numerical values for the parameters used in the simulation are lacking.

Does the zeta potential of the surface and the silica particles matter? How are the mobility parameters depending on the surface/material? The authors could comment on this, and thus clarifying the specificity of the chemical gradient used here compared to, say, an electrolyte gradient, a CO₂ gradient, or other systems used in the diffusiophoresis literature.

The authors assume h , the height of the simulated disc over the substrate, small. A numerical value or a measurement could be useful.

The assumption that a raft can be approximated to a disc may be questionable: the surface of a raft is approximated with a plane surface. In the case of the bottom surface, it faces a truly plane surface (the substrate) at a height h considered 'small'. However, the bottom of the raft is actually formed by the assembly of micrometer-sized particles and thus has micrometer-sized voids. Those voids are much deeper than h and not negligible compare to the diameter of the raft. In those voids the particle surface is exposed to the chemical gradient and the viscous drag from the diffusiophoresis. With this considered, I find it surprising that the simulation fits quantitatively. Any scaling factors involved to account for the geometrical approximation?

How often do particles assemble in more than one layer above the surface? It looks like this does happen looking at the movie in the SI and the images in the figures of the main text. How much does the experimental raft deviate from the assumed geometry? Does this matter?

Reading the conclusion of the manuscript about the competition of diffusiophoresis and -osmosis, a seemingly obvious experiment to challenge the model is to change the material of either the substrate or the passive particles. Can the author do the experiment or comment on why not.

Minor:

Non-exhaustive list of sentences in need of rephrasing, just for page 2: 'with blue light ... hematite', 'is sediment ...', 'experimental observed...'

Chamber is rectangular and the particles sediment to the bottom, but of which side? Is the observation chamber 100um high? Or 2mm high?

Scale bar in figure 3A is missing.

Reviewer #2 (Remarks to the Author):

Overall an interesting publication looking at the effects of phoretic and osmotic propulsion in active colloidal rafts, however I feel there are few points that need to be addressed:

Scale bar missing on C (3rd image) and D.

It would be good to state how many samples were considered in terms of experimental data?

When considering different ellipsoidal particles how did the amount of passive particles accumulating over illumination power and time differ? I guess it is partially shown by the error bars but it not clear how these have been generated? This also comes back down to how many samples were taken? Does the orientation of the ellipsoid make any difference?

I am wondering has temperature been taken into account, I might have missed it but as far as I can see both in the experimental design and the modelling temperature has not been considered? I would imagine this to have a strong effect in terms of assembly both time and size? How did illumination power affect the temperature of the experimental setup? This should be measured and confirmed to not alter the samples to ensure no temperature dependant relationship v.s. wavelength.

In terms of additives, has extra care been taken when selection H₂O₂ to ensure there are not interfering stabilising agents that could affect any propulsion etc?

It would be good to have a photo in SI to visualise the actual experimental setup.

Was there any bubble production within the closer capillary system? To ensure there are no interfering flow effects. There is a high chance of this occurring if you have high volume fractions of particles even if they are passive.

It would be good to do some MSD analysis of the passive as well as active particles alongside to see the differences. Similarly, it might be useful to also carry out persistence length or fractal dimension analysis techniques on passive and active particles.

I would also like to see the blank data against the reported data. This should be data with and without H₂O₂. As well as blank measurement with and without H₂O₂ with and without illumination over the given time. And supporting example movie files and figures similar to the one shown in Fig 1.

Was the light source across the whole sample or just a small area, as this much cause a temperature gradient within the sample and thus induce flow movements. This needs to be shown now to influence the experimental results.

Reviewer #3 (Remarks to the Author):

In this manuscript, the authors study the collective behavior of a quasi-two-dimensional mixture of active and passive colloidal particles using experiments, numerics and theory. Upon illumination, the active particles create chemical gradients around themselves which leads to a phoretic, non-reciprocal interaction between the active and passive particles and to their self-organization into a self-propelled colloidal raft, consisting of one active particle at the center surrounded by successive layers of passive particles. The raft's transport properties are experimentally characterized, and compared to numerical results obtained using a Brownian dynamics simulation. The authors find discrepancies between the experiments and numerics, which they attribute to the fact that the numerical simulations only take into account phoretic effects and disregard the presence of the substrate over which the colloids are located. The central claim of this manuscript is that the active particle induces an osmotic flow over the substrate by generating a chemical gradient close to it, which then exerts a force on large colloidal rafts. To support that claim, the authors develop a theoretical model in which they calculate the velocity of a raft-like disk in the presence of an osmotic flow using the Lorentz reciprocal theorem.

This is, as far as I know, the first work studying in detail the influence of the surface below a large self-propelled collective of active colloids on their collective motion. Furthermore, the authors used experimental, computational, and theoretical tools to characterize the system they study, yielding a detailed analysis of its behavior. Thanks to the novelty of the study and the variety of presented results, I believe this work will be of interest to the active colloids community.

However, I do have some major concerns regarding this work, which I think should be addressed before its publication. First, I am not sure that the data shown in the manuscript fully backs up the authors' claim that "the diffusioosmotic flow induced by the catalytic particle due to the near surface is necessary to describe the motion of active particles...". As mentioned by the authors, the osmotic contribution to equation (4) is indeed not necessary to explain the dependence of the raft velocity v_c as a function of the ratio χ / A . In particular, in Fig.4, the linear fit shown would be the same in the absence of osmotic effects, as proven by equation (36). Furthermore, while the self-propulsion velocity and MSD shown respectively in Fig. 2C and D are lower for the simulations than for the experiments, their plots as a function of time look similar. Could the observed difference between experiments and numerics then not be a matter of choosing the right simulation parameters? Additionally, if, as claimed at the end of the manuscript, "it is reasonable to assume that μ_S is comparable to μ_p , and have the same sign", shouldn't the velocity in the presence of osmotic effects then have a smaller magnitude than in their absence, unlike what is seen in Fig.2C? Could the authors also comment on why, despite being of the same material, the substrate and the colloids would have different diffusiophoretic mobilities? Overall, I would be more convinced about the relevance of osmotic effects if equations (4) and (5) were more strongly tied to the experimental and numerical data shown in the manuscript. In my opinion, these equations deserve a longer discussion, as they constitute an important result and the manuscript currently ends shortly after their introduction. If possible, I think the manuscript would also strongly benefit from additional results tying equation (5) to the experimental and numerical observations. For instance, could the simulations somehow take into account the presence of osmotic flow, e.g. as an additional force on the active particles? Or could the theoretical model capture the change in self-propulsion direction with the disk radius found in experiments? Could experiments be ran on a substrate with different properties than the one used in the manuscript, in order to observe differences in the raft velocity and MSD? While I do not think all these extensions are strictly necessary, the presence of at least some of them would make the work more cohesive and the role of osmotic effects clearer. If the authors think these additions are not currently doable or relevant, I would suggest at least highlighting the limits of their current approach and possible extensions to overcome them in the outlook portion of the conclusion.

My second main concern is that the Methods section of the manuscript seems insufficiently detailed, and needs changes and additions in order for the work to be fully reproducible. Those additions could either be made in the Methods or in the Supplementary Information.

Regarding the experimental portion of the work, the following methods are missing:

- measurement method of the surface area A of the colloidal raft
- raft velocity v_c and relative velocity of two colloids Δv_r (fig.2C) measurement methods
- measurement methods of the geometric quantities a , b (fig.3.A and B)
- measurement methods of the angle β in Fig.3C
- What does the distance of the red points to the center mean in Fig.3c? And how is the "non-linear regression" shown with the green continuous line calculated?

Regarding the numerical simulations, it was unclear to me exactly which parameters the Brownian dynamics simulations use. It would be useful to have a complete list of which simulations were ran accompanied by a table showing the corresponding parameters, including the simulation time step and the total simulation time / number of snapshots taken. An explanation of how these parameters were obtained would also be necessary: while it is the case for the characteristic colloid velocity v_0 , it is not clear to me how the value of τ_c is measured, and what exactly the chosen value of the ratio $\bar{\mu}$ is. The simulation shown in Supplementary Movie 3 is done "by imposing that the cluster move with the hematite at the rear." How is that achieved?

Here are a few more minor suggestions and remarks:

- 1) Shouldn't the expression of the characteristic velocity given in the Methods be $V_0 = \alpha_r \mu_a / (12 D_c)$, rather than $\alpha_r \sigma_a^2 / (12 D_c)$, which does not seem homogeneous to a velocity?
- 2) Should the sum in equation (12) should be over α , rather than $j \neq i$?
- 3) No definition for the gradient ∇_{\parallel} operator is given.
- 4) I could not find a definition of parameter ξ shown in equation (15).
- 5) Equation (5) is repeatedly mentioned as equation (35)

6) In the Methods, the descriptions of equations (11) (passive to active response) and (12) (active to passive

response) seem swapped to me.

As a final general point, in my opinion the manuscript lacks the level of polish one expects in papers published in Nature Communications. I have for instance found typos in equations (eqs. (13) and (14) have a parameter τ instead of τ_c , V instead of V_i in eq. (3), missing bracket in the average of d at the end of page 3...), as well as typos ("more than 100 hundred colloids" in the introduction), grammatical errors ("being d the distance traveled" at the end of page 3) and unclear formulations ("The corresponding heatmap of such field" should be "the heat map of the corresponding velocity field") in the text. Some terms are also explained too far from the equations they appear in (the v_0 in equation (3) should be immediately defined as a characteristic velocity) or not at all ($d\Omega$ is not defined as a surface element in (9), and the domain of integration is not explicitly given). I kindly encourage the authors to perform a thorough proof-reading of their manuscript before resubmission, as the points cited above negatively impact the comprehension and reception of their work.

I would overall recommend this manuscript underwent major revisions, focused on the main two points highlighted at the beginning of this review.

Response to the Reviewers' comments

We thank the Reviewers for their thorough reading of our manuscript and for the constructive suggestions/criticisms. Our point-by-point responses and the corresponding changes made to the manuscript are described below.

Response to Reviewer #1

Reviewer: *The manuscript..... self-propelling colloids. And so I can recommend the manuscript for publication.*

Authors: We thank the Reviewer for the positive comments and for resuming in a very detailed way our manuscript.

Reviewer: *The manuscript is well structured and figures are of high quality, however the manuscript would clearly benefit from proofreading. Also, the authors assume a lot when exposing the simulations. I would suggest revising the way the model is presented to make it more accessible, eventually recapitulating key elements of previous papers, and adding numerical values.*

Authors: We thank the Reviewer for finding our manuscript well-structured and appreciating the images. We have performed an exhaustive proofreading of it, correcting all typos and rephrasing it to make the message clearer. Moreover, we have also revised the way the model is presented to make it more accessible, as suggested by the Reviewer, recapitulating its key elements, and adding numerical values. The major changes in the manuscript have been highlighted in blue, while minor corrections to improve the clarity of the text have been implemented without colour.

Reviewer: *Major:*

It occurred to me quite late in the manuscript that the only chemical gradient considered in this system is the gradient of the reaction product oxygen. The gradient of reagents due to local depletion near the catalyst particle is neglected. This was first clear to me when reading ref 43. Even then, the argument for neglecting it is still unclear. Is that because $\text{grad } c / c_0$ is negligible? I think it needs to be clarified early in the ms.

Authors: We agree with the Reviewer that probably this aspect in the manuscript was not explained with enough details. We note that O_2 and H_2O_2 possess approximately the same diffusion coefficient in water, estimated to be about $\sim 2 \cdot 10^{-9} \text{ m}^2/\text{s}$ as described in [S. Kihara, et al. Biophys J. 106, 1882 (2014)] for the dioxygen and Ref. [A. O. Tjell et al. Sensing and Bio-Sensing Research 21, 35 (2018)] for the hydrogen peroxide. Given that the catalytic reaction produces one dioxygen molecule for every two hydrogen peroxide molecules consumed, we can infer that the gradients of both products are related within the entire environment by the equation:

$$\nabla\phi = \nabla[\text{O}_2] = -\frac{1}{2}\nabla[\text{H}_2\text{O}_2]$$

Rather than tracking the concentrations of both products throughout our equations, we have chosen to represent them both by the quantity ϕ , which equals the amount of oxygen produced. Consequently, the quantity μ , which determines the slip velocity along the surfaces, is effectively a mobility associated with the catalytic reaction, accounting for the depletion in hydrogen peroxide and the increase in oxygen, and it is adapted to the convention we have chosen for ϕ . We have commented this point in the main text, by writing on page 3, column 2 the following:

“We note here that the diffusion coefficients of dioxygen and hydrogen peroxide in water are approximately equal ($D_c = 2 \times 10^{-9} \text{ m}^2/\text{s}$) [New Reference 1, New Reference 2], therefore $\nabla \phi = \nabla [\text{O}_2] = -\frac{1}{2} \nabla [\text{H}_2\text{O}_2]$. Since only the concentration gradient matters for osmosis and phoresis, we consider only the quantity ϕ for simplicity. Consequently, the surface flow mobilities μ introduced in this study are associated with the gradients resulting from hydrogen peroxide decomposition, rather than from a specific chemical compound.”

[New Reference 1] S. Kihara, D. A. Hartzler, and S. Savikhin, Oxygen concentration inside a functioning photosynthetic cell, Biophys J. 106, 1882 (2014).

[New Reference 2] A. O. Tjell and K. Almdal, Diffusion rate of hydrogen peroxide through water-swelled polyurethane membranes, Sensing and Bio-Sensing Research 21, 35 (2018).

Reviewer: *How far does the concentration gradient extend around the catalytic particle? How is the length scale in figure 2B related to the maximum number of shells obtained for a given illumination power in figure 1D? Is the saturation of the raft size directly related to a gradient threshold reached at this distance from the catalytic particle? This is not clear.*

Authors: According to our model, the concentration should vary following a power law relationship with the distance, $1/r$ and the gradient as $1/r^2$ around the particle. This assumption is supported by the measurement of the pair interaction between an active and a passive particle. The size of the cluster corresponds to an equilibrium between the attractive phoretic force and the thermal fluctuation. When the osmotic force is too low, the thermal fluctuations would dominate and prevent the colloidal raft to grow further. We have explained this point in the text, on page 3, column 1 by writing:

“The saturation of the raft size corresponds to a balance between the effective phoretic potential energy and the thermal energy, of the order of $k_B T$, being k_B the Boltzmann constant and $T \sim 293 \text{ K}$ the thermodynamic temperature.”

Reviewer: *Since only the concentration of oxygen matters, could the author give the diffusion coefficient of oxygen used, the rate J , the mobilities etc.? Numerical values for the parameters used in the simulation are lacking.*

Authors: The specific values of the diffusion coefficient, and of the rate J are encapsulated within the rescaled mobility value, given by $\bar{\mu} v_0 = 11.6 \pm 0.4 \mu\text{m s}^{-1}$ which we extract from the experimental data using the fitting procedure detailed in the article. Giving the simplification to only use one concentration to describe the system, we should

only verify that the diffusion coefficient for oxygen and hydrogen peroxide are mostly equal, as we have described in our previous answer. Regarding the numerical values used for the simulations, we apologize if some of these values are missing within the text. We have placed all these values in a Table within the supporting Information and mention most of them in the Method section of the manuscript.

Reviewer: *Does the zeta potential of the surface and the silica particles matter? How are the mobility parameters depending on the surface/material? The authors could comment on this, and thus clarifying the specificity of the chemical gradient used here compared to, say, an electrolyte gradient, a CO₂ gradient, or other systems used in the diffusiophoresis literature.*

Authors: As the Reviewer correctly pointed out, the zeta potential of the surface of the materials in our system is important, since it prevents the colloidal particles from adhering to the surface through electrostatic repulsion. In particular, in our experiments we have employed a basic solution to increase the double layer around the particles and electrostatically stabilize them. However, considering the osmotic/phoretic effect, the zeta potential may not be significant, as the involved chemical products are neutral, as described for example in: S. Marbach et al., *Chemical Society Reviews* 48, 3102 (2019).

Considering the mobility parameters, preliminary investigations (see answer later) showed the important role of the material, particularly in modifying the substrate, which in turn affects the size of the clusters at steady state and its behaviour. In particular, we find that when using a polystyrene substrate, the raft is smaller and do not display self-propulsion. Establishing a direct correlation between these phenomena and the corresponding surface mobility modifications is a complex task. Additionally, changes in the pH of the solution have been observed to impact surface mobility, as reported in Palacci et al. *J. Am. Chem. Soc.* 135, 15978–15981 (2013). There, hematite particles were observed to repel each other at pH 6 on plastic substrates, while the interaction become attractive on a glass substrate. Understanding the chemistry and surface physics aspects falls beyond the scope of this paper.

Regarding the experiments reported in the diffusiophoretic literature, these typically involve well-controlled gradients of a single chemical specie, often salt, achieved through microfluidic devices. However, in our case, we are examining concentration gradients resulting from a chemical reaction, generated from a punctual source. To our knowledge, there are no well-controlled experiments measuring surface mobility specifically for the dissociation of hydrogen peroxide, despite its widespread use in the community of active particles.

Reviewer: *The authors assume h , the height of the simulated disc over the substrate, small. A numerical value or a measurement could be useful.*

Authors: Measuring the elevation h of the raft from the surface it is not a simple task given the fact that the raft itself is not transparent to visible light since the central hematite particle adsorb light. This prevents the use of optical techniques such as total internal reflection microscopy. However, we can obtain an approximate estimate by measuring the mean square displacement (MSD) of a hematite particle. The elevation h can be

extracted from the diffusion coefficient, using the Stokes Einstein relationship, and considering the correction due to the hydrodynamic interaction with the wall as introduced by Faxen (J. Happel and H. Brenner, Low Reynolds number hydrodynamics). We have performed these measurements, as also required by Reviewer #2, and add the corresponding MSD in the Supplementary Information. The measured diffusion coefficient of a passive hematite particle is $D_a = 0.16 \mu\text{m s}^{-1}$, from which we obtain an estimate of the surface elevation to be $h \sim 300 \text{ nm}$. We write this value in the text on page 2, column 1.

Reviewer: *The assumption that a raft can be approximated to a disc may be questionable: the surface of a raft is approximated with a plane surface. In the case of the bottom surface, it faces a truly plane surface (the substrate) at a height h considered 'small'. However, the bottom of the raft is actually formed by the assembly of micrometer-sized particles and thus has micrometer-sized voids. Those voids are much deeper than h and not negligible compare to the diameter of the raft. In those voids the particle surface is exposed to the chemical gradient and the viscous drag from the diffusioosmosis. With this considered, I find it surprising that the simulation fits quantitatively. Any scaling factors involved to account for the geometrical approximation?*

Authors: We agree with the Reviewer that the assumption of the model to consider the raft as a flat disk is indeed a strong approximation. However, the strength of our theoretical model lies in its simplicity, since it illustrates analytically the competition between phoresis and osmotic flow, which is the main point of this article. Replacing the disk model with a collection of spheres would significantly increase the complexity of the system, making impossible to develop analytical solutions.

Regarding the agreement with the numerical simulations, which indeed consider the spherical shape of the particles. We note that in Figures 2(a) and 2(b), the simulation parameters are obtained directly from the experimental measurements of the effective pair interactions. Essentially, the simulation can be viewed as an extension of the pair interactions to the multi-body interactions occurring within the colloidal raft. In a previous work [Ref. 54 of the manuscript], this approach provided a good quantitative depiction of the behaviour of mixtures composed of hematite and silica colloids. Thus, we expected that also in this case the simulations would be able to describe the experimental results. However, as shown by Figures 2(c,d) we find significant differences, which prompted us to explain the observed self-propulsion in term of the osmotic flow.

Reviewer: *How often do particle assemble in more than one layer above the surface? It looks like this does happen looking at the movie in the SI and the images in the figures of the main text. How much does the experimental raft deviate from the assumed geometry? Does this matter?*

Authors: Effectively, as the Referee point out, in some cases the passive silica particles can assemble above the raft due to the thermal fluctuations. However, based on our observations, this phenomenon is limited to a few colloids on top of the first layer. We have not encountered significant vertical stacking of colloids nor the formation of a 3D shells around the hematite particles. Considering the dynamics of the clusters, we believe that this phenomenon does not significantly impact its growth and propulsion. This is

supported by consistent cluster sizes observed across experiments and simulations, indicating that the system maintains a two-dimensional nature, preventing extensive stacking.

Reviewer: *Reading the conclusion of the manuscript about the competition of diffusiophoresis and -osmosis, a seemingly obvious experiment to challenge the model is to change the material of either the substrate or the passive particles. Can the author do the experiment or comment on why not.*

Authors: Effectively, as also required by Reviewer #3, we have performed further experiments by changing the nature of the substrate and demonstrating that the wall has a crucial effect in the raft formation and dynamics. We have performed the experiments on a polystyrene (PS) petri dish (Corning incorporated 430588). In contrast to the glass, on the PS substrate we find that the phoretic and osmotic effects dramatically decrease. Indeed, we observe that as soon as the blue light is turned on, a cluster start to form but it did not show any self-propulsion. Moreover, at parity of experimental conditions, the raft acquire a lower size than on the glass substrate, respectively reaching an area $A = 50 \mu\text{m}^2$ (PS) versus $A = 125 \mu\text{m}^2$ (glass). These measurements are reported in the image below which is also place in the Supporting Information.

Caption: Evolution over time of the cluster size for two light intensities and two different substrates. The blue curve corresponds to a light intensity $I = 125 \text{ mW cm}^{-2}$ and a glass substrate, the red one a light intensity $I = 60 \text{ mW cm}^{-2}$ and a glass substrate, and the blue curve a light intensity $I = 125 \text{ mW cm}^{-2}$ and a polystyrene substrate. For all experimental data the black lines are corresponding averages, and the shaded region denotes the confidence interval for $P = 0.95$.

These experiments make evident that changing the substrate has more influence on the osmotic effect, related to the flow along the substrate, than on the phoretic one, related to the flow along the particle surface. The hypothesis in our article suggests that the osmotic flow is stronger with a polystyrene substrate compared to a glass substrate. This explains why, at the steady state, the clusters are smaller, since such flow acts against clustering. Regarding the absence of motion above the PS substrate, it may appear

surprising that the clusters do not move despite the stronger osmotic flow. However, a divergent flow along the substrate implies that, due to incompressibility, an equally strong, vertical flow pressing the clusters against the wall. In the case of the PS, this force is strong enough to prevent the clusters from moving. We have commented these results in the main text, by writing on page 5, column 2 the following:

“Indeed, the importance of the proximity of the wall can be demonstrated by changing the nature of the substrate. In a separate set of experiments, we have repeated the assembly process above a polystyrene (PS) petri-dish. As shown in Supplementary Figure 3, we observe a decrease of the cluster area compared to the glass. Moreover, the assembled raft did not display any self-propulsion behaviour, sign that the PS influences more the osmotic effect, related to the flow along the substrate, rather than the phoretic one. Thus, the rafts that growth on PS are smaller. Moreover, a strong osmotic flow along the substrate implies the presence of a vertical flow pressing the clusters against the wall due to incompressibility. In the case of PS, this force is strong enough to prevent the clusters from moving.”

Reviewer: *Minor:*

Non-exhaustive list of sentences in need of rephrasing, just for page 2: ‘with blue light ... hematite’, ‘is sediment ...’, ‘experimental observed...’.

Authors: We thank the Reviewer for spotting these non-clear sentences. We have proofread the manuscript several times adjusting typos and correcting all unclear sentences including the one mentioned by the Reviewer.

Reviewer: *Chamber is rectangular and the particles sediment to the bottom, but of which side? Is the observation chamber 100um high? Or 2mm high?*

Authors: The experimental chamber has a rectangular shape, and it is characterized by an inner height of 100 micron and a lateral extension of 2mm. We apologize if this information was not clear in the manuscript, and we have corrected it both in the main text and in the Method section.

Reviewer: *Scale bar in figure 3A is missing.*

Authors: We have added the scale bar in Figure 3A and in Figure 1 as required also by Reviewer # 2.

Response to Reviewer #2

Reviewer: *Overall an interesting publication looking at the effects of phoretic and osmotic propulsion in active colloidal rafts, however I feel there are few points that need to be addressed:*

Authors: We thank the Reviewer for the positive feedback on our manuscript and the constructive criticisms, which we have all taken into account, please see below.

Reviewer: *Scale bar missing on C (3rd image) and D.*

Authors: We have added the scale bars in the corresponding images.

Reviewer: *It would be good to state how many samples were considered in terms of experimental data?*

Authors: We have followed the suggestion of the Reviewer and provide this information in the manuscript. The data illustrated in Figures 1(f), 2(c,d), 3(b), and 4 were generated from 12 different samples. In each experiment we have replicated the long exposure time to reach steady state conditions. For the Figure 1(e), each point has been obtained with at least independent 5 samples. For the Figure 2(b), from which simulation parameters were extracted, were used data from 64 different pairs of silica/hematite colloids. Additionally, for Figure 3(c), due to the rapid clustering, 18 shorter experiments were conducted in addition to the 12 ones made over long time to accumulate enough statistics for the smallest clusters. We specify these values within the main text and figure captions.

Reviewer: *When considering different ellipsoidal particles how did the amount of passive particles accumulating over illumination power and time differ? I guess it is partially shown by the error bars but it not clear how these have been generated?*

Authors: During the experiments with long exposure time, the smallest cluster observed covered a surface area of $100 \mu\text{m}^2$, while the largest one an area of $170 \mu\text{m}^2$. The error bars were derived assuming a Student's t-distribution, indicating a confidence level with a probability of $P = 0.95$ that the mean lies within them. We have specified this point in the caption of Figure 2

Reviewer: *This also comes back down to how many samples were taken? Does the orientation of the ellipsoid make any difference?*

Authors: In this case, 12 different experiments were examined. Initially, we hypothesized that the orientation of the ellipsoid might influence the direction of the raft motion. This hypothesis was tested by analysing the angle $\tilde{\alpha}$ between the short axis of the ellipsoid and the velocity vector. As illustrated in the figure below, the statistical distribution of this vector appears uniform within the error bar.

Caption: Probability distribution function of the angle $\tilde{\alpha}$, formed between the short axis and the velocity vector of a colloidal raft.

This suggests that the orientation of the ellipsoid has a negligible effect on the direction of motion. Similarly, in studying clustering behaviour, we did not observe any correlation between the cluster area and the ellipsoid orientation. Consequently, our conclusion is that the shape of the ellipsoid does not exert a noticeable influence. We have commented this point in the main text, by writing on page 4, column 1 the following:

“The iso-velocity lines are slightly elliptic, instead of circular as expected for an isotropic system. However, we find that this anisotropy is rather weak, and do not affect the raft dynamics. As shown in Supplementary Figure 2, the probability distribution function for the angle $\tilde{\alpha}$, formed between the short axis of the hematite particle and the velocity vector v_c of the raft, is flat within the error bars.”

and add the corresponding image as supplementary figure 2.

Reviewer: *I am wondering has temperature been taken into account, I might have missed it but as far as I can see both in the experimental design and the modelling temperature has not been considered?*

Authors: In our experiments the temperature influences the particle diffusion via thermal fluctuations. Indeed, the size of the raft is given by the balance between an effective attractive phoretic potential and the thermal energy of the order of $k_B T$. We have measured the eventual presence of temperature changes as a function of the light power and find that it is negligible in our system as explained in the next answer. Thus, in our experiments the thermodynamic temperature is constant over all time, we also give its value in the main text. In the numerical simulations, the temperature is considered as standard white noise delta correlated, and effectively it led to a finite cluster size: when the phoretic attraction is comparable or lower than $k_B T$ the passive particles detach from the clusters. Finally, in our theoretical model the temperature is implicitly considered in the release rate of the solvent and in the diffusion rate.

Reviewer: *I would imagine this to have a strong effect in terms of assembly both time and size? How did illumination power affect the temperature of the experimental setup? This should be measured and confirmed to not alter the samples to ensure no temperature dependant relationship v.s. wavelength.*

Authors: The question of the Reviewer on the eventual change of temperature as a function of power light trigger us to perform further experiments in order to exclude this effect. Directly measuring the temperature in our system is not possible since the colloidal suspension is hermetically sealed within the observation chamber and introducing any sensor will inevitably alter the colloidal dynamics by introducing impurities. However, we can indirectly measure the effect of temperature by monitoring how and if the particle diffusion coefficient changes when increasing the light power at constant wavelength (note that in all our experiments we did not change the wavelength). We have measured the mean square displacement of different samples made of passive silica particles with

3-micron diameters. We choose this larger size since in the long time tracking these particles display negligible out of place fluctuations with respect to the silica particle with 1 micron size. The results are shown in the image below that we also place in the supporting information (supplementary figure 5).

Caption: (A) One dimensional mean square displacements (MSDs) as a function of time of passive silica particles with 2 microns diameter and subjected to blue light at different light intensities I . For $I = 125 \text{ mW cm}^{-2}$ the MSDs along the two orthogonal directions (x, y) are shown to demonstrate the isotropic diffusion and the absence of net drift. All the curves display standard diffusive dynamics and overlap, sign of the absence of any temperature variation. (B) Corresponding diffusion coefficient (D) as a function of the light intensity (I) extracted from the MSDs from 10 different particles and averaged along the two orthogonal directions (x, y) at each light intensity. The experimental data display small fluctuations within the error bars around a mean value of $D = 0.109 \pm 0.002 \text{ } \mu\text{m}^2 \text{ s}^{-1}$.

We comment this pointy in the method section, by writing:

“We have independently checked the effect of the illumination power on the eventual variation of the temperature within the experimental system or the presence of drifts due to light absorption. As shown in Supplementary figure 5, we have tracked the position of silica spheres with 2 micron diameters at different applied powers, and measured the MSDs from which we extract the corresponding diffusion coefficients. We find that, in all cases, the MSDs are diffusive and isotropic along the two orthogonal directions in the particle plane (x, y).”

Reviewer: *In terms of additives, has extra care been taken when selection H₂O₂ to ensure there are not interfering stabilising agents that could affect any propulsion etc?*

Authors: We have careful checked the reference of the H₂O₂ used for the experiments (H₂O₂ 30% in water, Fisher Scientific - BP2633-500) and we didn't find any stabilising agent added to the solution of hydrogen peroxide.

Reviewer: *It would be good to have a photo in SI to visualise the actual experimental setup.*

Authors: We have followed the suggestion of the Reviewer and add three images in the Supporting Information file showing the experimental setup and the experimental cell. We have mentioned these images in the main text as Supplementary Figure 1.

Reviewer: *Was there any bubble production within the closer capillary system? To ensure there are no interfering flow effects. There is a high chance of this occurring if you have high volume fractions of particles even if they are passive.*

Authors: Before being introduced into the rectangular glass cell via capillarity, the colloidal dispersion undergoes a one-minute treatment in an ultrasound bath. This process effectively removes any trapped bubbles formed during preparation. Additionally, we took care to significantly reduce the quantity of hematite in the solution. This step serves to mitigate both bubble-related issues and the likelihood of cluster collisions during the long-time experiments.

Reviewer: *It would be good to do some MSD analysis of the passive as well as active particles alongside to see the differences. Similarly, it might be useful to also carry out persistence length or fractal dimension analysis techniques on passive and active particles.*

Authors: We have added to the Supplementary Material the measured mean square displacements as a function of time for the silica colloids and for the hematite particles with and without light, from which we extract the corresponding diffusion coefficients. We have also added the mean square angular displacement of the hematite, which helps extracting the rotational diffusion coefficient.

Caption: (a) Translation mean square displacement of a passive silica sphere with 1 μm diameter and (b) of a hematite particle in absence of light (black square) and with blue light (blue circles). (c) Angular mean square displacement of the hematite particle in absence (black square) and presence (blue circles) of light. In each graph the slopes that indicate diffusive ($\sim t$) and ballistic ($\sim t^2$) regimes are shown in red.

We have commented the measured diffusion coefficients in the main text. Regarding the analysis of the persistence length l_p , it was not possible to extract l_p for the passive particles since they display passive diffusive dynamics for all times. In contrast, from the analysis of the active hematite particles we find a persistence length of 1.8 microns for the hematite particle in the presence of light and H_2O_2 . We mention these results in the main text.

Reviewer: *I would also like to see the blank data against the reported data. This should be data with and without H₂O₂. As well as blank measurement with and without H₂O₂ with and without illumination over the given time. And supporting example movie files and figures similar to the one shown in Fig 1.*

Authors: We followed the Referee suggestion, and performed further experiments to measure whether in absence of light or in absence of H₂O₂ the hematite particles are able to attract the passive silica spheres and to grow clusters. We use similar experimental conditions as the one shown in Figure 1 in terms of concentration of H₂O₂ and light intensity. In all cases, we find that cluster can be grown only when the active and passive colloids are in a mixture of water and H₂O₂ and subjected to light. We add the corresponding videos in the supplementary information:

- **Supplementary Movie 2:** This videoclip illustrates the dynamics of a hematite particle and silica microspheres (1 micron diameter) dispersed in a hydrogen peroxide water solution and in absence of any illumination. Both particles display standard diffusive dynamics without being affected by any phoretic flow.
- **Supplementary Movie 3:** In this videoclip we show the steady state dynamics (after ~10 min of illumination) and the absence of attraction between an hematite particle and the silica microspheres (1 micron diameter) when dispersed in pure water (no Hydrogen peroxide) and subjected to the strongest power illumination, $I = 125 \text{ mW cm}^{-2}$. Also in this case, these particles display standard diffusive dynamics without being affected by any phoretic flow.

We comment on these blank measurements in the main text, on page 3, column 1 by writing the following:

“We note that the raft formation can only be obtained via the phoretic flow, induced by the photoactivated decomposition of hydrogen peroxide in water. Indeed, in a separate set of experiments, we have checked that in absence of light (Supplementary Movie 2) or in absence of hydrogen peroxide under blue light (Supplementary Movie 3) both the hematite and the silica particles display simple diffusive dynamics without any sign of phoretic attraction.”

Reviewer: *Was the light source across the whole sample or just a small area, as this much cause a temperature gradient within the sample and thus induce flow movements. This needs to be shown now to influence the experimental results.*

Authors: Considering the entire sample, only a small portion of it is illuminated. At the scale of the field of view, which is considerably smaller than the entire sample (as shown in Figure 1(c)), the illumination is uniform. We have checked the presence of any flow or drift due to temperature gradient but, as explained in our previous answer, we do not observe it. Indeed, except for the hematite in very small quantity, the sample is mostly transparent, which exclude any gradient resulting from light adsorption.

Response to Reviewer #3

Reviewer: *In this manuscript, the ... using the Lorentz reciprocal theorem.*

This is, as far as I know, the first work studying in detail the influence of the surface below a large self-propelled collective of active colloids on their collective motion. Furthermore, the authors used experimental, computational, and theoretical tools to characterize the system they study, yielding a detailed analysis of its behavior. Thanks to the novelty of the study and the variety of presented results, I believe this work will be of interest to the active colloids community.

Authors: We thank the Reviewer for the positive comments on our manuscript and for finding this work of interest to the active colloidal community.

Reviewer: *However, I do have some major concerns regarding this work, which I think should be addressed before its publication. First, I am not sure that the data shown in the manuscript fully backs up the authors' claim that "the diffusioosmotic flow induced by the catalytic particle due to the near surface is necessary to describe the motion of active particles...". As mentioned by the authors, the osmotic contribution to equation (4) is indeed not necessary to explain the dependence of the raft velocity v_c as a function of the ratio χ / A . In particular, in Fig.4, the linear fit shown would be the same in the absence of osmotic effects, as proven by equation (36).*

Authors: As highlighted by the Reviewer, a linear behaviour is also anticipated in a purely diffusiophoretic system. However, the primary objective of Figure 4 is to demonstrate that, despite the model's simplicity, it accurately predicts the variations in velocity as a function of the raft size and asymmetry. In fact, the compelling argument in favour of considering the osmotic flow lies in the inability to reconcile both clustering and the direction of cluster motion with a purely phoretic system. On one hand, the observation of clustering implies that the mobility of passive particle $\mu_p < 0$, as otherwise, the hematite would repel the silica colloids rather than attract them. On the other hand, assuming a purely phoretic phenomenon would imply that the raft velocity v_c is negative, meaning that the cluster moves with the hematite at the front. This would be consistent with what we observed for the smallest clusters but not for the larger raft. On the other hand, a cluster moving with the colloids at the rear would require $\mu_p > 0$, assuming a purely diffusio-phoretic phenomenon, which contradicts the observed effect.

The simplest approach to resolve this discrepancy is to consider the role of osmotic flow in the problem. This flow is often overlooked in the community of self-propelled particles but is well-known in the microfluidic community that may induce an opposing effect and compete with diffusiophoresis.

Reviewer: *Furthermore, while the self-propulsion velocity and MSD shown respectively in Fig. 2C and D are lower for the simulations than for the experiments, their plots as a function of time look similar. Could the observed difference between experiments and numerics then not be a matter of choosing the right simulation parameters?*

Authors: The simulation parameters that allow direct comparison with the experimental data have been extracted from the measurements. Eq. (3) is fitted to the experiments, to obtain the characteristic velocity $v_0\bar{\mu}=11.6 \mu\text{m/s}$. This fit to the experimental data shows that the second term of Eq. 3 has a negligible contribution to the velocity in experiments. Thus, we are in a situation where $\bar{\mu} \gg 1$, and the dynamics are dominated by the passive diffusiophoretic mobility. We apologize if in our previous version of the text we did not clearly mention this effect. We now emphasise in the “Numerical simulations” section by writing:

“We find the second term of the rhs in Eq. 3 to be negligible to fit the expression...”

As we write in the section “*Details of the numerical simulation*”, we used two parameters to write the equation in dimensionless variables. These are the diameter of the passive particles, σ_p from which we rescale the position and the characteristic time. To define the latter, we use the velocity contribution that was determined previously from the experiments, $\tau_c = \sigma_p / V_0 \bar{\mu}$. Using the experimental value of σ_p we determine the characteristic time as $\tau_c \sim 0.086$ s. With these parameters we can write the dimensionless output of the simulation back to dimensions in corresponding physical units. Besides $\bar{\mu}$, there are only three additional parameters which relate to the excluded volume interactions, the spring joining the two beads constituting the active particle, and the Péclet numbers due to the Brownian diffusion. All these parameters are chosen according to experimental results or observations. We have expanded the section “*Details of the numerical simulation*” to state more clearly where these parameters come from in the simulations, and how we choose each of them.

Reviewer: *Additionally, if, as claimed at the end of the manuscript, "it is reasonable to assume that μ_S is comparable to μ_p , and have the same sign", shouldn't the velocity in the presence of osmotic effects then have a smaller magnitude than in their absence, unlike what is seen in Fig.2C?*

Authors: Several arguments can justify why the velocity of the cluster is lower in the simulation than in the experiments. A first argument is that in the simulations, the interaction between the particles reproduces the pair interactions (hematite - silica) without any other particle nearby, while in the experiment many other colloids are around, occupying a region of space which has not been considered in the simulations. This may induce a local higher concentration gradient than expected, and therefore a higher velocity. Another argument is that the simulations have been implemented assuming a purely phoretic system, they have been calibrated with experiments, in which occur a competition between osmosis and phoresis. If the diffusiophoresis dominates just slightly the pair interaction, it is possible that the simulations generalising the pair interaction to a whole cluster might result in a lower velocity than in the experiments for which the osmosis is drastically more dominant.

Reviewer: *Could the authors also comment on why, despite being of the same material, the substrate and the colloids would have different diffusiophoretic mobilities?*

Authors: In reality, the colloids and the substrate are not made of the same material. Indeed, the colloidal raft is composed of a central hematite particle which is surrounded

by several shells of silica colloids. The latter are synthesized via the Stöber technique and are made usually of amorphous silicon. In contrast, the substrate above which the raft move is made of a transparent, borosilicate glass which is obtained via a different synthetic process. The presence of silica in both materials may lead to think that the osmotic flow mobility is probably similar, however the different synthetic process to obtain these materials and their different internal structure (amorphous versus crystalline) implies that the mobility should not be equal.

Reviewer: *Overall, I would be more convinced about the relevance of osmotic effects if equations (4) and (5) were more strongly tied to the experimental and numerical data shown in the manuscript. In my opinion, these equations deserve a longer discussion, as they constitute an important result and the manuscript currently ends shortly after their introduction.*

Authors: we agree with the Reviewer that equations 4 and 5 (now Equations 5 and 6) were shortly commented in the text. Thus, we have added a longer discussion on these equations and the corresponding implications. In particular, we write on page 6 and 7 the following:

“Moreover, our approach provides also the dependency on $\frac{\chi}{A}$ for the disk velocity that agree well with experimental observations, as shown in Fig.4.

A characteristic feature of our system is the symmetric competition between osmotic and phoretic effects. The symmetry between both contributions arises directly from the short distance h relative to the cluster radius a . Firstly, we can neglect the slight variation in concentration distribution between the disk bottom and the substrate surface facing it, resulting in identical slip velocities modulo the mobility factor. Secondly and more importantly, in the dual problem, this thin-walled geometry induces a shear flow between the disk and the substrate, resulting in symmetric but opposite viscous stresses on the two surfaces. Since the distribution of the viscous stress of the dual surface weights the slip flow contribution in the expression of the osmotic/phoretic force, this feature is the origin of the symmetric competition between osmotic and phoretic forces for large clusters.

Intriguingly, if we consider a scenario where the condition $h/a \ll 1$ is less satisfied, such as when the disk size becomes comparable to the distance h , the simple shear flow might disappear in the dual problem. Consequently, the viscous stress on the bottom of the disk becomes larger than that on the substrate, breaking the symmetry of the competition in favour of the phoretic force. This explains the observed transition in the dominant force between osmosis and phoresis for small to large clusters.”

Reviewer: *If possible, I think the manuscript would also strongly benefit from additional results tying equation (5) to the experimental and numerical observations. For instance, could the simulations somehow take into account the presence of osmotic flow, e.g. as an additional force on the active particles?*

Authors: In our manuscript, the role of the numerical simulations is to highlight the outcome of the dynamics in a purely diffusio-phoretic system. By comparing the

simulations and the experimental results, we find that diffusiophoresis alone, which was used in previous works in the field, cannot reproduce the dynamics of the raft. Thus, we have developed a theoretical model that demonstrates the contribution of the osmotic flow to the raft dynamics with an additional term, which can predict the experimentally observed change of the direction of motion of the raft. We agree that more involved numerical simulation schemes could have been used to capture the osmotic flow effect, with the use for example of propagators that would mimic the slip velocity on the solid surface. Thus, we believe that the suggestion of the Reviewer is quite valid and interesting and could be a valid starting point for further research, however it will go beyond the message of the current manuscript.

Reviewer: *Or could the theoretical model capture the change in self-propulsion direction with the disk radius found in experiments?*

Authors: With the current model it is possible to extract a critical radius a_c marking the change between the self-propulsion direction by considering the competition between the forces applying on the lower and upper face of the disk. However, this approach would give $a_c \sim h$ which is ridiculously small, and it is not consistent with what we observed experimentally. This discrepancy, probably come from the fact that we neglect the thickness and the complex composition of the cluster and that it should disturbed the pure shear flow near the cluster edges.

Reviewer: *Could experiments be ran on a substrate with different properties than the one used in the manuscript, in order to observe differences in the raft velocity and MSD? While I do not think all these extensions are strictly necessary, the presence of at least some of them would make the work more cohesive and the role of osmotic effects clearer. If the authors think these additions are not currently doable or relevant, I would suggest at least highlighting the limits of their current approach and possible extensions to overcome them in the outlook portion of the conclusion.*

Authors: The question raised by the Reviewer prompted us to perform further experiments by changing the nature of the underlying substrate. This question was raised also by Reviewer #2, where we have provided further experimental data on the raft assembly on a polystyrene substrate, and we have described the corresponding results in the main text. Please refer to the corresponding answer to Reviewer #1.

Reviewer: *My second main concern is that the Methods section of the manuscript seems insufficiently detailed, and needs changes and additions in order for the work to be fully reproducible. Those additions could either be made in the Methods or in the Supplementary Information.*

Authors: We apologize if the description of the experiments/simulations in the Method section was insufficient. We have followed the suggestions of the Reviewer and rewritten most of these parts adding further information both in the Method and Supporting Information. Below we detail our main changes following the comments of the Reviewer.

Reviewer: *Regarding the experimental portion of the work, the following methods are missing:*

- measurement method of the surface area A of the colloidal raft
- raft velocity v_c and relative velocity of two colloids Δv_r (fig.2C) measurement methods
- measurement methods of the geometric quantities a , b (fig.3.A and B)
- measurement methods of the angle β in Fig.3C

Authors: We have followed the suggestions of the Reviewer and add the detailed description of these missing methods as supplementary information text 2.

Reviewer: - *What does the distance of the red points to the center mean in Fig.3c? And how is the "non-linear regression" shown with the green continuous line calculated?*

Authors: Figure 3(c) depicts in polar coordinates the probability distribution function of the angle β which is the angle between the velocity vector of the cluster and the asymmetry vector b . The latter connects the geometric center of the cluster with the location of the hematite within the cluster. Thus, this angle gives information on the propulsion direction: $\beta = 0^\circ$ ($\beta = 180^\circ$) means that the raft propels with the active particle at the front (rear). The distance of the red points from the center gives the amplitude of the distribution. The green continuous line is a fit assuming a wrapped normal distribution with one or two peaks. We specify this point in the caption of Figure 3.

Reviewer: *Regarding the numerical simulations, it was unclear to me exactly which parameters the Brownian dynamics simulations use. It would be useful to have a complete list of which simulations were ran accompanied by a table showing the corresponding parameters, including the simulation time step and the total simulation time / number of snapshots taken. An explanation of how these parameters were obtained would also be necessary: while it is the case for the characteristic colloid velocity v_0 , it is not clear to me how the value of τ_c is measured, and what exactly the chosen value of the ratio $\bar{\mu}$ is. The simulation shown in Supplementary Movie 3 is done "by imposing that the cluster move with the hematite at the rear." How is that achieved?*

Authors: The simulations contained 700 passive particles, and one active particle, at the same area fraction used in experiments, $\varphi = 0.06$. In order to determine the Pe number in the simulations, we computed the diffusion coefficients of the active (D_a) and passive (D_p) particles from the MSD. The obtained value of Pe is within the range [40-70] which is used in the numerical simulations. Simulations at different Pe inside this range showed that the behaviour is qualitatively the same, while the precise Pe number quantitatively changes the number of shells around the active particle. Thus, we have chosen a value of the Pe number that resulted in a number of shells similar to that of the experiments. The results shown in the manuscript correspond to 40 simulations with the same parameters, with different random initialisations, which served to test the sensitivity of the results to the initial position of the particles. The qualitative results were the same, so the statistical analysis was performed over all these 40 simulations.

The total simulation time correspond to $t = 5220$ s, comparable to that of the experiments, measuring every 0.13 s, and we use a time step $dt = 4.35 \cdot 10^{-5}$ s. We set the mobility ratio to $\bar{\mu} = 10$. At this value, this parameter is compatible with the experimental results, and the term $1/r^2$ dominates the cluster growth and dynamics. Increasing $\bar{\mu}$ leads to

quantitatively the same results. Smaller values of $\bar{\mu}$ results in an incompatibility of fitting the experimental curve using Eq. 3 of the main text. We summarise the parameters used in the following table which is also provided in the Supplementary file:

Name	Symbol	Value used
Number of active particles (dumbbells)	N_a	1
Number of passive particles	N_p	700
Area fraction of particles	A_φ	0.06
Diameter of passive particle	σ_p	$1\mu\text{m}$
Diameter of active particle in dumbbell	σ_a	$1.3\mu\text{m}$
Spring constant of active dumbbell	k	100
Rest length of the spring constant	l_0	$0.5\mu\text{m}$
Péclet of active particles	Pe_a	60
Péclet of passive particles	Pe_p	53.8
Ratio between mobilities	$\bar{\mu}$	10
Rescaled characteristic velocity	$v_0\bar{\mu}$	$11.6 \mu\text{ m s}^{-1}$
Characteristic time scale	τ_c	0.086 s
Total simulation time	t_{tot}	5160 s
Dimensionless ratio	$\mu_a k \sigma_p / v_0 \bar{\mu}$	50
Dimensionless ratio	$\epsilon_{a,p} \mu_{a,p} / v_0 \bar{\mu}$	10

We have expanded the section “*Details of the numerical simulations*” and explained in detail each term in the dynamic equation and how these parameters were chosen. The expansion of this section now clarifies all the parameters used in the simulations, and how they were selected. In particular, we write the following:

“We choose the dimensionless spring parameter, $\mu_a k \sigma_p / v_0 \bar{\mu} = 50$ so that the spring joining the two beads is rigid and almost incompressible, resulting in a rigid active particle. “

and

“We choose $\mu_a \epsilon_{a\beta} / v_0 \bar{\mu} = 5$, for any combination of active and passive interactions, to simulate the excluded volume interactions. The system dynamics are dominated by $Pe_{a,p}$ which are the Péclet number of the active and passive particles resp., and by the mobility ratio $\bar{\mu}$. Note that in the dimensionless units, from Eq. 13 a large value of $\bar{\mu}$ implies a small contribution of the velocity induced by passive particles on the active one. This is indeed the situation observed in the experiments. In the simulations, we give to $\bar{\mu}$ a value which is compatible with that obtained from fit in Fig.2(B), $\bar{\mu} \sim 10$. Around this value, we performed additional simulations, to determine

whether small deviations may affect the cluster formation. We observed that the MSD and cluster growth of the active raft did not change quantitatively. Finally, to determine the Péclet numbers, we use the diffusion coefficients extracted from the experiments (D_p , D_a). Thus, we use the values $Pe_a = 53.8$ and $Pe_p = 60$, such that they fulfil the relationship, $Pe_a / Pe_p = D_p / D_a$. We performed 24 simulations including $N_p = 700$ passive particles and $N_a = 1$ active dumbbell, each simulation performed with different random initial positions of all particles. We use the same area fraction as in the experiments, $A_\phi = 0.06$. The total simulation time for each run is $t_{tot} = 60000\tau_D = 5160$ s, snapshots for analysis are taken every $t_f = 1.4\tau_D = 0.12$ s, and the time step is $dt = 5 \cdot 10^{-4}\tau_c = 4.35 \cdot 10^{-5}$ s. All the parameters used for the numerical simulations are provided in Supplementary Table 1.”

As for the measurement of the characteristic time τ_c , we now state more clearly in the manuscript that this time depends on the particle diameter σ_p which is known in the experiments and on the characteristic velocity $v_0\bar{\mu}$ which is also obtained from the experimental data. We also noticed that there was a typo in the previous value of τ_c . The computed characteristic time is $\tau_c = 0.086$ s, as we write now in the text.

Reviewer: *The simulation shown in Supplementary Movie 3 is done "by imposing that the cluster move with the hematite at the rear." How is that achieved?*

This is achieved by computing the velocity of the centre of mass of the active raft in the simulations, and redistributing the velocities inside the cluster such that the average centre of mass velocity is now inverted, while keeping the attraction between the passive and active particles the same away from the raft. We clarify this point in the text by writing on page 5, column 1:

“To confirm this hypothesis, we have implemented a specific simulation where we invert the average velocity of the center of mass of the raft at each time step, while keeping the active-passive interactions outside the cluster following Eq. 3. In this case, as shown in Supplementary Movie 5, we find that the cluster moves with the hematite at the rear, and observe a much longer persistence length, closer to the experimental results.”

Reviewer: *Here are a few more minor suggestions and remarks:*

1) *Shouldn't the expression of the characteristic velocity given in the Methods be $V_0 = \alpha_r \mu_a / (12 D_c)$, rather than $\alpha_r \sigma_a^2 / (12 D_c)$, which does not seem homogeneous to a velocity?*

Authors: We thank the reviewer for noticing this typo which we have corrected in the Methods.

Reviewer: 2) *Should the sum in equation (12) should be over α , rather than $\sum_{i \neq j}$?*

Authors: We thank the reviewer for noticing this. The subindex α refers to each particle composing the active dumbbell, while i and j subindexes refer either to a passive or to an active particle.

Referee: 3) *No definition for the gradient ∇_{\parallel} operator is given.*

Authors: We define ∇_{\parallel} to be the derivative tangential to the surface of a sphere. We have introduced this definition in the text, on page 3, column 2.

Reviewer: 4) I could not find a definition of parameter ξ shown in equation (15).

Authors: The term ξ corresponds to the stochastic process that accounts for the thermal fluctuations. After writing equations 13 and 14 in dimensionless variables, $\hat{\xi}$ corresponds to a Gaussian distribution centered at zero, and with variance unity. We have updated the derivation and accounted for the average force that particles feel due to the fluctuations in this term, by means of calculating $\langle \hat{\xi}^2 \rangle$. We have also further on how this estimation of the cluster radius is obtained. We add in the Method section (page 9) the following:

“The cluster radius a can be estimated at steady state, when the diffusiophoretic interaction becomes comparable to the averaged forces due to thermal noise. Comparing the diffusiophoretic term from Eq. 14 and the diffusion one Eq. 15 we get

$$a \sim \frac{\sigma_a}{\sqrt{\langle \hat{\xi}^2 \rangle}} \left(\frac{Pe_p}{2} \right)^{1/4} .$$

Using the experimental value of $Pe_p=60$, and approximating the hematite particle as a spherical one with an equivalent radius of $\sigma_a \sim 1.55 \mu\text{m}$, we obtain $a \sim 9 \mu\text{m}$, in agreement with numerical simulations. Thus, the growth rate of the cluster is compatible with an attraction $1/r^2$. Note that the diffusion coefficient D_p controls the size of the raft. This result has been confirmed in the simulations by varying Pe_p inside the range compatible with the experimental measurements.”

Reviewer: 5) Equation (5) is repeatedly mentioned as equation (35)

Authors: We thank the Reviewer for pointing out this inconsistency. We refer now to the corresponding equation in the main text, and to the equation in the section with the details of the model when appropriate.

Reviewer: 6) In the Methods, the descriptions of equations (11) (passive to active response) and (12) (active to passive response) seem swapped to me.

Authors: We have checked equations (11) and (12) and they are not swapped. The passive particle moves towards an active particle following $1/r^2$, while an active particle moves to the passive one following $1/r^5$ according to the theoretical derivation. However, we noted we swapped the description in the text, just before expressions (11) and (12). We thank the reviewer for noticing this incongruence which we correct in the new version.

Reviewer: As a final general point, in my opinion the manuscript lacks the level of polish one expects in papers published in Nature Communications. I have for instance found

typos in equations (eqs. (13) and (14) have a parameter τ instead of τ_c , V instead of V_i in eq. (3), missing bracket in the average of d at the end of page 3...), as well as typos ("more than 100 hundred colloids" in the introduction), grammatical errors ("being d the distance traveled" at the end of page 3) and unclear formulations ("The corresponding heatmap of such field" should be "the heat map of the corresponding velocity field") in the text.

Authors: We have revised the manuscript correcting all typos and performing a careful proofreading of the text to improve and clarify the language.

Reviewer: *Some terms are also explained too far from the equations they appear in (the v_0 in equation (3) should be immediately defined as a characteristic velocity) or not at all ($d\Omega$ is not defined as a surface element in (9), and the domain of integration is not explicitly given). I kindly encourage the authors to perform a thorough proof-reading of their manuscript before resubmission, as the points cited above negatively impact the comprehension and reception of their work.*

Authors: We follow the suggestion of the Reviewer and perform a thorough proof-reading of their manuscript. We hope that after this major revision, the manuscript has improved in clarity.

Reviewer: *I would overall recommend this manuscript underwent major revisions, focused on the main two points highlighted at the beginning of this review.*

Authors: We have followed all the suggestion of the Reviewer and improved the manuscript accordingly. We have addressed the two points highlighted by the Reviewer. We hope that after this revision the manuscript can be further considered for publication.

REVIEWERS' COMMENTS

Reviewer #1 (Remarks to the Author):

The authors have addressed my initial comments and I can recommend the manuscript for publication.

Reviewer #2 (Remarks to the Author):

Dear Authors,

Overall, I am happy with the amendments made to the manuscript.

Two final small points:

In terms of the oxygen bubbles randomly forming in the sample during the experiment, I am unsure that sonication is the best option to remove these only. I would have recommended to ensure the dissolved oxygen content of the solution is low such as with nitrogen bubbling of the solution prior to the experiment. But as long as no bubbled were there to induce any drift it should be fine.

I would only recommend to add some scale bars to the videos as well as a real time clock.

Reviewer #3 (Remarks to the Author):

The questions and concerns I had voiced in my initial review have been addressed by the authors to my satisfaction, and I only have a few extra minor points to make at this stage:

- Some of the phrasing in the new paragraph added at the end of page 5, regarding the experiments done with a polystyrene (PS) surface, is a bit confusing to me. How is the absence of self-propulsion a sign that PS influences osmotic more than phoretic effects? I would imagine that the phoretic effects only depend on the particle properties, and that changing the substrate from glass to PS does not influence them.

- If I am not mistaken, equation (18) is obtained from the diffusiophoretic and noise terms of equation (14), not of equations (14) and (15). Additionally, is τ the same as τ_c in equations (14) and (15)?

- In supplementary figure 3, the green curve is designated as blue

- Supplementary figure 6 (b): the first power law line should be t , not $t^{1/2}$, in order for the MSD to be diffusive as indicated in the caption

Given that most of my remarks concern small typos, I encourage the authors to perform an additional proofreading of their manuscript, focused on the material they have added as a response to the reviewers.

I would recommend the publication of this work in Nature Communications once the few points I have raised here have been addressed.

Response to the Reviewers' comments

We thank the Reviewers for reading again our manuscript and for the positive feedback. Our point-by-point responses and the corresponding changes made to the manuscript are described below.

Response to Reviewer #1

Reviewer: *The authors have addressed my initial comments and I can recommend the manuscript for publication.*

Authors: We thank the Reviewer for recommending our work for publication.

Response to Reviewer #2

Reviewer: *Overall, I am happy with the amendments made to the manuscript.*

Two final small points:

In terms of the oxygen bubbles randomly forming in the sample during the experiment, I am unsure that sonication is the best option to remove these only. I would have recommended to ensure the dissolved oxygen content of the solution is low such as with nitrogen bubbling of the solution prior to the experiment. But as long as no bubbles were there to induce any drift it should be fine.

Authors: We thank the Reviewer for this suggestion. Effectively nitrogen bubbling the solution is an alternative way to reduce the oxygen bubble due to the particle chemical reactions. However, we find that this was not required in our system due to the very low concentration of active particles that we use to generate the colloidal raft.

Reviewer: *I would only recommend to add some scale bars to the videos as well as a real time clock.*

Authors: We have followed the suggestion of the reviewer and added the scale bars and the clock to all the experimental videos in the supporting Information. The simulation videos (Supplementary Movie 4 and Supplementary Movie 5) already provide a scale in micron on their side and are in real time.

Response to Reviewer #3

Reviewer: *The questions and concerns I had voiced in my initial review have been addressed by the authors to my satisfaction,*

Authors: We thank the Reviewer for recognizing that we have been to address all the concerns/question raised.

Reviewer: *and I only have a few extra minor points to make at this stage:*

- Some of the phrasing in the new paragraph added at the end of page 5, regarding the experiments done with a polystyrene (PS) surface, is a bit confusing to me. How is the absence of self-propulsion a sign that PS influences osmotic more than phoretic effects? I would imagine that the phoretic effects only depend on the particle properties, and that changing the substrate from glass to PS does not influence them.

Authors: We agree with the Reviewer that the phrase at the end of page 5 may be a bit confusing. In this phrases one may mix the concept of competition between phoresis and osmosis, and the fact that changing the substrate should only affect the osmosis, which are different. To make it clearer, we have rephrased the sentence as follows:

“Moreover, the assembled raft did not exhibit any self-propulsion behavior, indicating that the modification of the osmotic effect due to the PS substrate also influences the raft’s mobility. Specifically, a radial osmotic flow around the hematite and along the substrate develops, in opposition to clustering, promoting a vertical flow pressing the clusters against the wall due to incompressibility. In the case of PS, this force is strong enough to prevent the clusters from moving.”

Reviewer: - If I am not mistaken, equation (18) is obtained from the diffusiophoretic and noise terms of equation (14), not of equations (14) and (15). Additionally, is τ the same as τ_c in equations (14) and (15)?

Authors: We estimate the size of the raft by comparing the strength of the attraction which comes from Eq. 14 with the strength of the noise that changes the particle orientation, which is described in Eq. 15. Indeed, ξ_{α} in Eq. 14 denotes the thermal noise but the term ξ in Eq.15 is also a thermal noise but that affect the particle orientation. Probably the confusion arises from the used notation since in Eq.18 the correct dimensionless variable $\hat{\xi}$ features, while this is not the case in Eqs. 14 and 15. We have corrected Eqs. 14 and 15, and write down $\hat{\xi}_{\alpha}$ and $\hat{\xi}$, respectively.

Finally, we agree with the Referee that τ is the same as τ_c , and we have corrected this in the main text.

Reviewer: - In supplementary figure 3, the green curve is designated as blue

Authors: We thank the Reviewer for spotting this mistake, which we have corrected in the new Supporting Information.

Reviewer: - Supplementary figure 6 (b): the first power law line should be t , not $t^{1/2}$, in order for the MSD to be diffusive as indicated in the caption

Authors: We thank the Reviewer for spotting this mistake, which we have corrected in the new version of the Supporting Information.

Reviewer: Given that most of my remarks concern small typos, I encourage the authors to perform an additional proofreading of their manuscript, focused on the material they have added as a response to the reviewers.

Authors: We have followed the suggestion of the Reviewer and performed an additional proofreading of the manuscript making special attention to the material added during the previous round of review.

Reviewer: *I would recommend the publication of this work in Nature Communications once the few points I have raised here have been addressed.*

Authors: We have followed all suggestions of the Reviewer and we hope that with these changes the manuscript can be accepted for its publication.